# Bayesian meta-analysis reveals the mechanistic role of slow oscillation-spindle coupling in sleep-dependent memory consolidation

**Thea Ng[1,2], Eunsol Noh[3], Rebecca MC Spencer[3,4,5]***

[1]Neuroscience & Behavior Program, Mount Holyoke College, South Hadley, United States; [2]Department of Mathematics & Statistics, Mount Holyoke College, South Hadley, United States; [3]Neuroscience & Behavior Program, University of Massachusetts, Amherst, United States; [4]Department of Psychological & Brain Sciences, University of Massachusetts, Amherst, United States; [5]Institute of Applied Life Sciences, University of Massachusetts, Amherst, United States

**\*For correspondence:**
rspencer@umass.edu

**Competing interest:** The authors declare that no competing interests exist.

## eLife Assessment

This **important** study presents a meta-analysis confirming a statistically significant association between slow oscillation-spindle coupling and memory formation, although the reported effects are limited (~0.5% of variance). The evidence is overall **convincing**, but the statistical methods may be difficult to follow for readers unfamiliar with advanced techniques. This work will be of particular interest to neuroscientists studying the neural mechanisms of sleep and memory.

**Abstract** The active system consolidation theory suggests that information transfer between the hippocampus and cortex during sleep underlies memory consolidation in humans. Neural oscillations during sleep, including the temporal coupling between slow oscillations (SO) and sleep spindles (SP), may play a mechanistic role in memory consolidation. However, differences in analytical approaches and the presence of physiological and behavioral moderators have led to inconsistent conclusions. This meta-analysis, comprising 23 studies and 297 effect sizes, focused on four standard phase-amplitude coupling measures including coupling phase, strength, percentage, and SP amplitude, and their relationship with memory retention. We developed a standardized approach to incorporate non-normal circular-linear correlations. We found strong evidence supporting that precise and strong SO-fast SP coupling in the frontal lobe predicts memory consolidation. The strength of this association is mediated by memory type, aging, and spatiotemporal features, including SP frequency and cortical topography. In conclusion, SO-fast SP coupling should be considered as a general physiological mechanism for memory consolidation.

## Introduction

Over the past three decades, accumulating evidence supports the role of sleep neural oscillations and their cross-frequency coupling in the spatiotemporal coordination across brain regions, supporting sleep-dependent memory consolidation (*Buzsáki and Draguhn, 2004*; *Hyafil et al., 2015*; *Klinzing et al., 2019*). During nREM sleep, oscillatory activities including cortical slow oscillations (SO; 0.16–4 Hz, *Figure 1A*), thalamocortical sleep spindles (SP; 8–16 Hz), and hippocampal sharp-wave

**Figure 1.** Measurement of phase-amplitude coupling (PAC) in slow oscillation and spindle events. (**A**) The origin of neural oscillations during sleep. *SO* slow oscillation, *SP* sleep spindle, *SWR* sharp wave ripple. In each subgraph, the vertical line indicates the typical amplitude of that sleep wave, while the horizontal line indicates the typical frequency and duration. Note that SPs also propagate along the cortex, and the figure only displays the origin. (**B**) Electrophysiology representation diagram of the SO-SP coupling. SO and SP amplitudes are normalized. The phase of SOs when SPs are at their maximum instantaneous amplitude is recorded as the coupling phase. The occurrence of SPs and SO-SP coupling is not necessarily continuous as shown in the diagram. (**C**) Coupling preferred phase and strength diagram. (Left) The phase and strength used in the circular plot are simulated data from existing dataset for visualization purposes only. At the group level, the mean circular direction shows the preferred SO phase, while the mean vector length shows the strength of the precise coupling. (Right) The phase of SO peaks is noted as 0, while the phase of SO troughs is noted as $\pm\pi$.

ripples (SWR; 80–300 Hz), along with their long-distance coordination, are widely believed to be closely associated with the process of transferring temporary encoded memory traces to cortical networks for consolidation (*Clemens et al., 2007*; *Maingret et al., 2016*; *Staresina et al., 2015*).

Given that non-invasive recordings (e.g.; EEG) cannot accurately detect SWRs in deep brain structures, electrophysiological studies in humans primarily focus on the role of SOs and SPs in memory consolidation. Consistent evidence indicates that, following intensive learning, SP density and its co-occurrence with SOs significantly increase compared to baseline nights and control groups (*Gais et al., 2002*; *Mölle et al., 2009*; *Mölle et al., 2004*; *Schmidt et al., 2006*; *Solano et al., 2022*). Correspondingly, measures of SPs and their coupling with SOs predict over-sleep retention performance of newly acquired memories (*Holz et al., 2012*; *Nicolas et al., 2022*; *Kumral et al., 2023*; *Kurdziel et al., 2013*; *Rodheim et al., 2023*).

The role of sleep oscillations in memory consolidation is supported by consistent neurobiological foundations. SOs are generated and dominant in the prefrontal cortex during slow-wave sleep (SWS) and propagate anteriorly to posteriorly as traveling waves (*Massimini et al., 2004*; *Kurth et al., 2017*; *Malerba et al., 2019*; *Niethard et al., 2018*). The corticothalamic input of SOs during their depolarization up-state phase organizes the synchronous occurrence of SPs in the thalamic reticular nucleus. Subsequently, SPs propagate back widely to cortical areas through synchronized thalamocortical projections, resulting in EEG-measured SPs (*Contreras et al., 1997*; *Kim et al., 1995*; *Marshall*

*et al., 2006*; *Neske, 2015*; *Oyanedel et al., 2020*; *Steriade, 2003*). The peak discharge of SPs in the cortex is associated with increased dendritic Ca²⁺ synchronization (*Niethard et al., 2018*; *Rosanova and Ulrich, 2005*; *Seibt et al., 2017*). This precise association enhances synaptic plasticity, leading to long-term changes in synaptic connections between cortical neurons, a mechanism for memory consolidation (*Lindemann et al., 2016*; *Martin et al., 2000*; *Miyamoto et al., 2017*; *Steriade and Timofeev, 2003*).

Despite extensive research on the functions of these oscillations during sleep and their causal sequences, it is unclear how the spatiotemporal coordination mechanisms facilitate specific temporal sequences of the peak discharge and the associated memory consolidation. Neuronal activity exhibits various forms of cross-frequency coupling, mainly including phase-phase coupling (PPC) and phase-amplitude coupling (PAC). The coupling is considered crucial in regulating the integration of information across multiple spatial and temporal scales (*Canolty and Knight, 2010*; *Palva et al., 2005*). Studies related to memory often focus on the unidirectional PAC regarding the strong amplitude modulation of faster oscillations (FO) driven by the phase modulation of relatively slower oscillations in a hierarchical order. The phase of SOs influences rhythmic spikes of FOs near the up-state peak or down-state trough of SOs (*Canolty et al., 2006*; *Fell and Axmacher, 2011*; *Jensen and Colgin, 2007*). In memory networks, PAC is considered a crucial mechanism supporting neural communication and plasticity (*Fell and Axmacher, 2011*).

The active system consolidation theory supports the subdivision of memory consolidation processes into three coordinated steps, which occur in the precise hierarchical temporal structure of SO-FO coupling involving SOs, SPs, and SWRs (*Born and Wilhelm, 2012*; *Takehara-Nishiuchi, 2021*; *Winocur and Moscovitch, 2011*): (1) Hippocampal SWRs driven by cortical SOs facilitate the repeated replay of newly encoded memories. (2) The maximum amplitude of SWRs precisely locks with the trough phase of SPs during the depolarized up-state of SOs (SP-SWR coupling). (3) The maximum amplitude of SPs couples with the up-state of SOs (SO-SP coupling). These phase-locked coupling dynamics are considered integral to consistent communication between the hippocampus and neocortex, constituting a key mechanism for the reactivation and redistribution of temporary memory traces across cortical areas (*Diekelmann and Born, 2010*; *Helfrich et al., 2019*).

Pharmacological and stimulation research has provided evidence for causal relationships between coupling and memory consolidation. For example, the increasing level of inhibitory neurotransmission driven by GABAergic drug zolpidem improves the phase precision and strength of SO-SP coupling and, in turn, contributes to enhanced memory retention performance (*Carbone et al., 2021*; *Kersanté et al., 2023*; *Niknazar et al., 2015*; *Zhang et al., 2020*). Studies using calcium imaging and SP stimulation explained the significance of the precise coupling phase for synaptic plasticity: SP spike discharges extracted through electrical stimulation during SO up-states efficiently modify excitatory neocortical synapses (*Rosanova and Ulrich, 2005*). Recently, *Niethard et al., 2018* observed that only SP spikes occurring around upstate peaks of SOs were accompanied by amplified calcium activity patterns to optimize synaptic plasticity. This spatiotemporal mechanism represents a critical perspective in the study of SO-SP coupling in memory consolidation.

When quantifying cross-frequency coupling, the spike-timing-dependent transfer theory (*Diekelmann and Born, 2010*; *Rasch and Born, 2013*) signified the importance of coupling phases and inconsistency in FO amplitudes. The coupling phase has been widely shown to reflect dynamic functional configurations, indicating remote communication and precise timing changes in synaptic activity among neural populations sharing cognitive functions (*Jiang et al., 2015*; *Sauseng and Klimesch, 2008*). Comparing to the depolarization state (positive half-wave) of cortical SOs which drives FOs including SPs, its inhibitory hyperpolarization state exhibits relative silence of neurons (*Neske, 2015*; *Mölle et al., 2002*). The preferred phase represents the phase of SOs at the maximum amplitude of FOs (*Figure 1B and C*). Another common method is the peri-event time histogram (PETH), representing the proportion of FOs centered (i.e. maximum trough) at different SO phase bins (*Kurz et al., 2021*; *Muehlroth et al., 2019*).

Phase alone does not reveal the inconsistent strength of FO amplitude between coupled and non-coupled SO phases within each time window. Therefore, the analysis of phase-amplitude distribution constitutes the so-called coupling strength, also defined as coupling inconsistency. In memory research, two commonly used quantification methods for coupling strength include mean vector length (*Canolty et al., 2006*; MVL) and modulation index (*Tort et al., 2008*; MI). Both determine

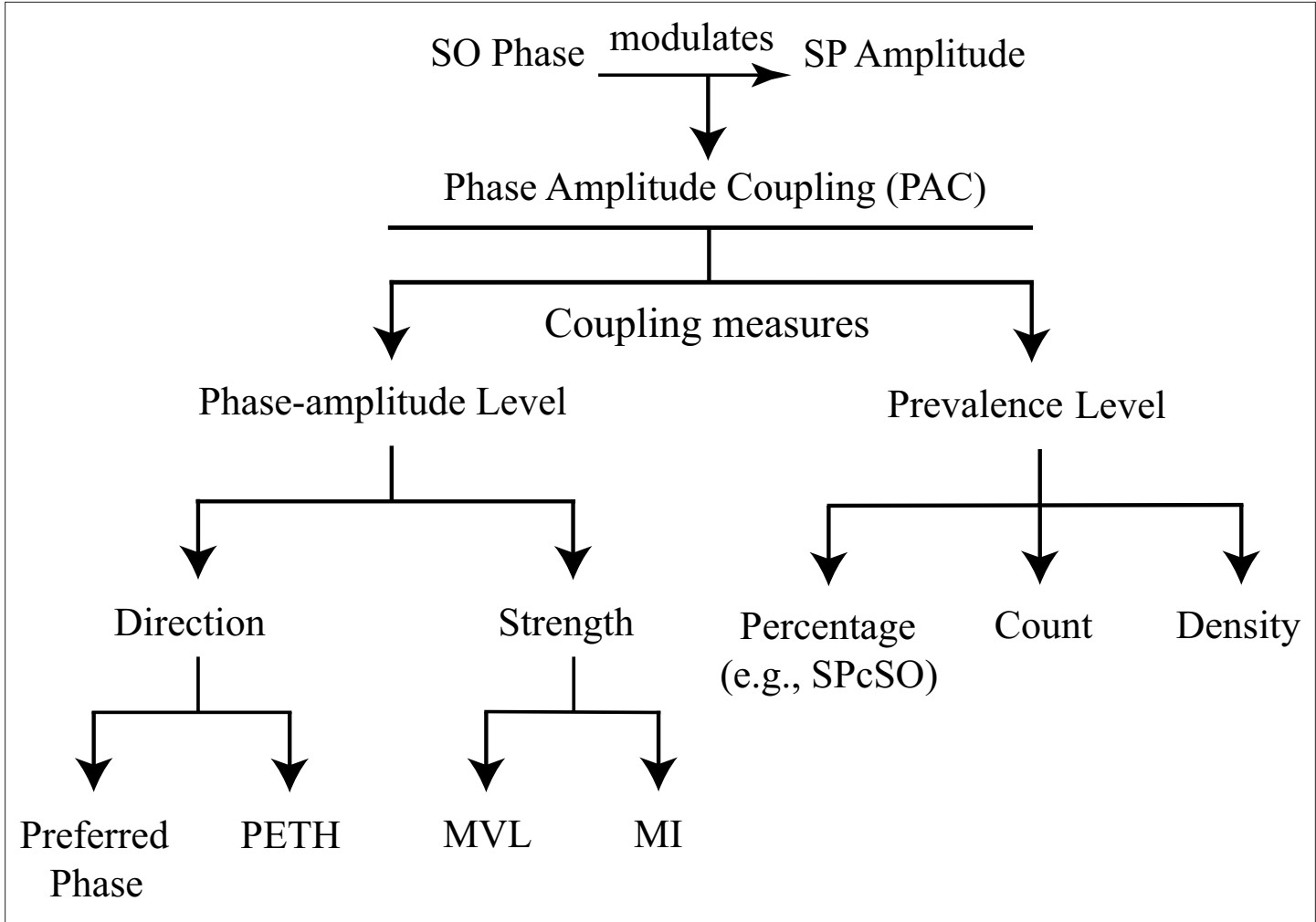

**Figure 2.** Hierarchical diagram of coupling measures. *PETH* peri-event time histogram, *MVL* mean vector length, *MI* modulation index. *SPcSO* Percentage of SPs coupled with SOs in all SP events. In contrast to the term 'frequency' used throughout the text in reference to the neural oscillation, the coupling prevalence in the diagram indicates the occurrence of SO–SP coupling events.

the strength of PAC by measuring the uneven distribution of FO amplitude in different SO phase directions. Past simulation studies have proved the effectiveness of these two methods in extracting coupling strength under different conditions and the existence of noise (*Hülsemann et al., 2019*; *Samiee and Baillet, 2017*; *Tort et al., 2010*).

Compared to the coupling phase and strength, which reflect the contrast on the phase-amplitude level, recent research also focuses on the overall occurrence of coupling events across the night. This involves assessing the prevalence of co-occurrence, including the coupling percentage (*Denis et al., 2021*; *Hahn et al., 2020*; *Halonen et al., 2021*; *Kurz et al., 2021*) (i.e. co-occurrence rate, % coupling events / oscillation events), coupling counts (*Niknazar et al., 2015*), and the coupling density (*Mylonas et al., 2020*). It is reasonable to categorize all measures related to co-occurrence under a category defined as 'coupling prevalence', in contrast to the phase and strength (see *Figure 2*).

In SO-SP coupling studies, although most of them supported the importance of coupling phase in predicting memory retention, this association varies in magnitude across memory types, sleep stages, and age (*Bastian et al., 2022*; *Cox et al., 2018*; *Denis et al., 2021*; *Donnelly et al., 2022*; *Hahn et al., 2020*; *Halonen et al., 2021*; *Halonen et al., 2022*; *Helfrich et al., 2018*; *Kurz et al., 2021*; *Mikutta et al., 2019*; *Weiner et al., 2024*). In contrast, the significance of coupling strength and percentage on memory consolidation is inconsistent across studies (*Kurz et al., 2023*; *Mikutta et al., 2019*; *Weiner et al., 2024*; *Hahn et al., 2022*).

Furthermore, previous meta-analyses of SP function consistently indicate that SP amplitude (and power) is the most effective predictor of memory consolidation and cognitive abilities (*Kumral et al., 2023*; *Ujma, 2021*). The measure of neural oscillation amplitude is reported based on changes compared to mean amplitude, and the absolute amplitude difference between peak and trough reflects the intensity of synchronized neural activity near that specific scalp region (*Teplan, 2002*). Although SP amplitude is not a direct measure of SO–SP coupling, and prior studies report discrepancies regarding whether its maximum value predicts coupling phase and strength (*Baena et al., 2023*; *Roebber et al., 2022*), evidence consistently indicates that the magnitude of SP amplitude is systematically modulated by SO phase, and this phase-dependent modulation constitutes a core mechanism of coupling (*Helfrich et al., 2018*; *Klinzing et al., 2019*; *Staresina et al., 2015*). Including SP amplitude reflects group differences in SP activity overnight and allows comparison of the roles of coupling metrics and SP alone, as examined in many of the included studies (*Kurz et al., 2021*; *Niknazar et al., 2015*; *Ladenbauer et al., 2021*). Also, only 4 studies reported SP amplitude separately for coupled events, which limits targeted analyses. Therefore, we also include the mean peak-to-trough amplitude and power of all SP events in the meta-analysis (*Figure 1B*).

Electrophysiology evidence from studies included in our meta-analysis (*Denis et al., 2021*; *Hahn et al., 2020*; *Mylonas et al., 2020*) and others (*Rodheim et al., 2023*; *Muehlroth et al., 2019*; *Bartsch et al., 2019*) reported that the association between memory consolidation and SO-SP coupling is influenced by a variety of behavioral and physiological factors under different conditions. Among the moderators that have received significant attention in recent research are memory types, development and aging, pharmacological manipulations, disorders, and sleep stages. The early dual-process hypothesis proposed a dissociation between hippocampus-dependent and non-dependent memories reinforced during SWS and REM periods, respectively (*Rasch and Born, 2013*): The standard active consolidation theory supported the role of hippocampal-cortical nesting during nREM sleep for the consolidation of hippocampus-dependent memories, including verbal, visual, and spatial declarative memories that are episodic-related (*Rasch and Born, 2013*; *Gais and Born, 2004*; *Marshall and Born, 2007*). In contrast, some research has linked the processing of non-hippocampus-dependent memory, such as emotional and procedural memories, with REM sleep (*Groch et al., 2013*; *Plihal and Born, 1997*; *Sara, 2017*; *Wagner et al., 2001*).

This outdated theory may have led early SO-SP coupling studies to focus primarily on the experimental design of declarative memory inferences. Recent updates to this foundational theory propose that the hippocampus plays a crucial role in the consolidation of non-hippocampus-dependent memories during nREM sleep through the reactivation of spatiotemporal contextual features (*Ackermann and Rasch, 2014*; *Boutin and Doyon, 2020*; *King et al., 2017*; *Sawangjit et al., 2018*; *Spencer et al., 2006*). This has sparked discussions and research on whether hippocampal involvement in SO-SP-SWR coupling constitutes a general physiological principle in all types of memory, or at least in memories with spatiotemporal contexts. Particularly in the association between SO-SP coupling and non-hippocampal-dependent consolidation, recent studies have obtained conflicting results (*Solano et al., 2022*; *Nicolas et al., 2022*; *Kumral et al., 2023*; *Cox et al., 2018*; *Mikutta et al., 2019*; *Mylonas et al., 2020*; *Hahn et al., 2022*; *Nishida and Walker, 2007*; *Wei et al., 2018*).

In addition, early behavioral experiments have demonstrated the sensitivity of sleep-dependent memory consolidation to age-related changes in the brain, from development to aging (*Kopasz et al., 2010*; *Spencer et al., 2007*). The most significant changes of sleep neural oscillations during development include the maturation and dominance of frontal-originated global SOs and frontal-detected SPs (*Hahn et al., 2019*; *Shinomiya et al., 1999*; *Timofeev et al., 2020*). This change is associated with the improvement of memory consolidation from childhood to adolescence. In older adults, the time spent in SWS sharply decreases, followed by a reduction in the number and amplitude of SOs (*Harand et al., 2012*; *Petit et al., 2004*). Their prefrontal SP events decrease by over 40% compared to young adults and are correlated with their weakened memory performance (*Mander et al., 2013*; *Martin et al., 2013*).

Taking a step further, *Helfrich et al., 2018* and *Hahn et al., 2020* have respectively identified the modulating roles of aging and development in the frontal SO-SP coupling, including changes in the precision of coupling phase, shifts in coupling topography, as well as improvements and impairments in memory retention performance. Without exception, all age-related studies on sleep-dependent memory consolidation emphasize the involvement of the prefrontal cortex and posterior

hippocampus, which contribute to episodic memory processing and consolidation. Differences in their gray matter volumes, and changes in structural integrity during development and aging, are related to the representation of neural oscillations (*Mander et al., 2013*; *Ishii et al., 2018*; *Saletin et al., 2013*).

Besides these moderators, overlooked in most studies is the region-specificity of different memory types and the frequency of SPs (*Geva-Sagiv and Nir, 2019*). The topography and frequency distribution coupling are neglected or subject to misleading interpretations when correlating other measures. SP frequency has traditionally been defined between 12 and 16 Hz (*Rechtschaffen, 1968*). However, recent research has found that SPs of different frequencies dominate at different phases of the SO cycle and in distinct cortical areas (*Mölle et al., 2011a*; *Zeitlhofer et al., 1997*). Therefore, studies started to analyze the SO-SP coupling by splitting SPs into fast and slow subtypes (*Solano et al., 2022*; *Cox et al., 2018*; *Perrault et al., 2019*). Since some studies found that fast SPs predominate in the centroparietal region, while slow SPs are more common in the frontal region, a significant amount of studies selectively extracted specific types of SPs from limited electrodes (*Perrault et al., 2019*; *Schreiner et al., 2021*; *Dehnavi et al., 2021*). Some studies even averaged all electrodes in their spectral and/or time-series analysis to estimate metrics of oscillations and their couplings (*Nicolas et al., 2022*; *Mölle and Born, 2011b*; *Denis et al., 2022*). This narrowed measure falls into the pitfall of considering all SPs as the regional oscillation and overlooking the out-of-phase distribution of SOs in different cortical areas (*Nir et al., 2011*).

Acknowledging the existence of regional SO and SP events, recent evidence supports that global SO and SPs, as traveling waves, constitute the majority of their oscillation cycles (*Massimini et al., 2004*; *Dickey et al., 2021*; *Muller et al., 2016*). Traveling waves are considered to play a crucial role in the transmission of information, including memory, between specific brain areas (*Muller et al., 2018*). Once projected onto the cortex, global SPs exhibit rotational characteristics, looping from the temporal lobe to the parietal and frontal lobes sequentially (TPF) and then travel back to the temporal lobe (*Muller et al., 2016*; *O'Reilly and Nielsen, 2014*). This results in phase gradients in the burst of SP spikes across different cortical areas, which represents the direction of asymmetric cortical propagation and might explain the phase shifts in coupling (*Ermentrout and Kleinfeld, 2001*; *Hindriks et al., 2014*). It is worth noting that in other types of oscillations, the traveling waves have been found to also follow descending frequency gradients (*Zhang et al., 2018*).

Dynamic spatiotemporal features of global SPs are believed to create necessary conditions for the synaptic plasticity (*Dickey et al., 2021*). Recently, relevant clues have emerged in research on SO-SP coupling. *Hahn et al., 2020* and *Helfrich et al., 2018* found that only the phase and strength of coupling detected in frontal electrodes had sufficient predictive power for the success of memory retention. *Denis et al., 2021* reported a phase gradient from anterior to posterior area in SO-fast SP coupling by channels. Although all this evidence indicates the necessity for investigation of the spatiotemporal specificity of SO-SP coupling, current between-study differences and the ambiguity in defining SP types (see Discussion: Challenges of current statistical approaches in measuring EEG-behavior associations) introduce uncertainty to the measure of coupling topography. To our knowledge, there is no previous study that has systematically analyzed the dynamic spatiotemporal distribution of SO-SP coupling and its association with the frequency of oscillations and memory consolidation.

In summary, current research on the SO-SP coupling and memory consolidation faces multiple methodological challenges. Disadvantages of polysomnographic (PSG) studies include long recording times, limitations on subject recruitment, and variations in signal processing approaches. Researchers have the flexibility to choose analytical methods and selectively report results that favor their hypotheses (*Cordi and Rasch, 2021*; *Reverberi et al., 2020*), thereby increasing the risk of false positives and lack of reproducibility. Considering methodological limitations, varied analytical approaches, disparate quality of reports, and conflicting evidence, there is a clear urgency for a meta-analysis to consolidate the true effect sizes reported in studies, as well as propose standardized and targeted research and analysis methods.

To incorporate a broader range of studies without increasing between-study heterogeneity and publication bias, we requested unreported or selectively omitted effect sizes, as well as individual-level coupling data that were pre-processed but not included in correlation analysis. This approach helped us include more studies adopting a similar experimental design but using incomparable analysis methods, which is common in memory research. In addition, these data provide us with sufficient

statistical power to compare spatiotemporal shifts in coupling phase and its association with memory consolidation.

The purpose of the current meta-analysis and review is to aggregate studies of SO-SP coupling to understand inconsistent results across studies. To be more specific, we used Bayesian hierarchical models to examine whether the cross-frequency coupling between SOs and SPs is associated with memory consolidation and determine which specific measure(s) of SO-SP coupling most accurately predict memory retention performance. In addition, we conducted moderator analyses to understand if physiological and behavioral factors modulate the strength of the relationship between SO-SP coupling and memory consolidation. We propose five questions for moderators: (1) Is SO-SP coupling a general physiological mechanism for memory consolidation or targeted at specific type(s) of memory?; (2) How do development and aging affect the association between coupling and memory consolidation?; (3) Is the coupling detected under specific (functional) region(s) more indicative of the memory retention?; (4) In which frequency range(s) of SPs is coupling more likely to be linked to memory consolidation?; (5) Does the stage and bout of sleep affect the measure of coupling-memory association? Besides the main study, we also analyzed the preferred distribution of coupling phases under different cortical areas and SP frequencies. In the rest of the discussion, we focus on summarizing conflicts and misinterpretations in current research, proposing strategies for research standardization, as well as discussing the implications of our results for future research focus and clinical applications.

## Results

### Study characteristics

In the final dataset of 23 studies included in the data analysis, the participants had an average age of 24.8 years, ranging from 7.3 to 78.0. The average female representation among participants was 49.5%. The average sample size for each study was 31.7 participants, ranging from 10 to 151. Each study contributed four effect sizes for each coupling measure on average. The included studies consist

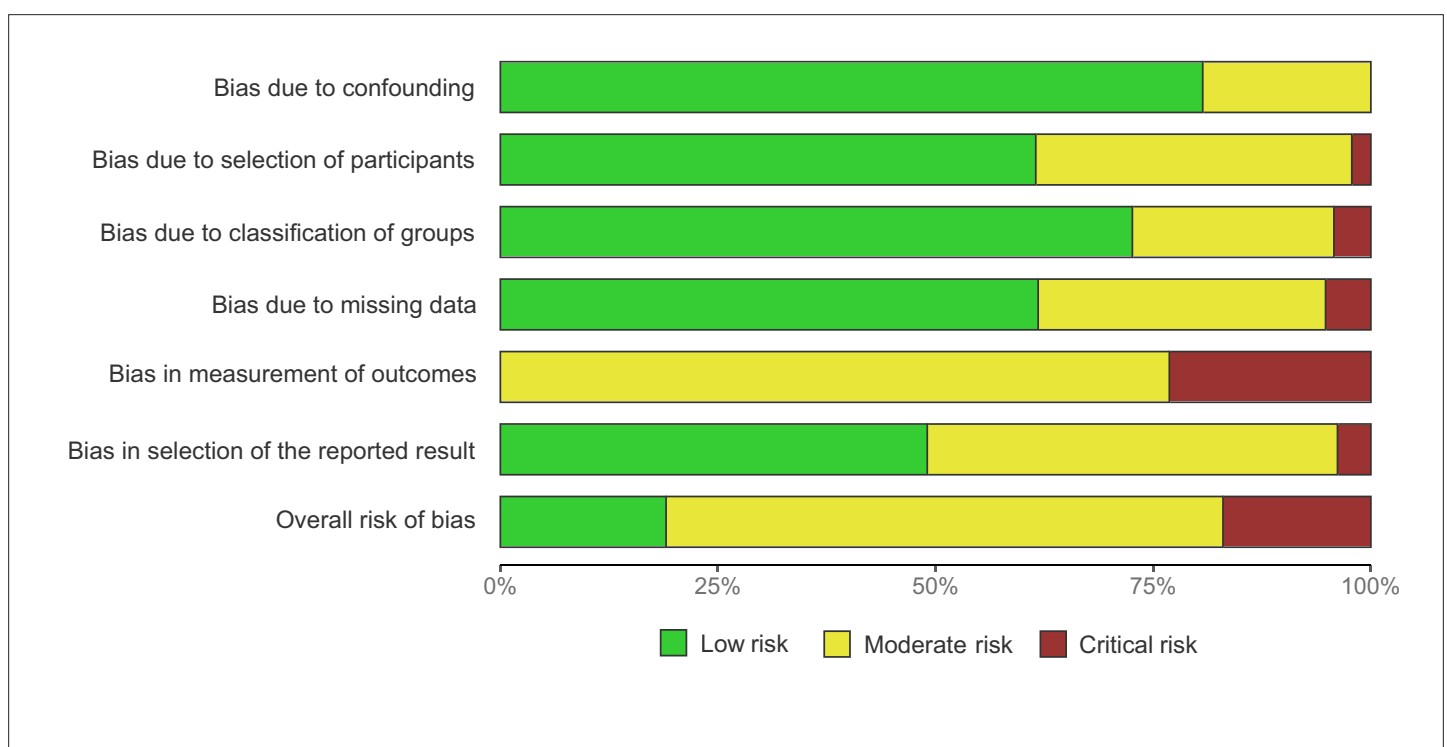

**Figure 3.** Risk of bias assessment summary plot adapted from ROBINS-I (**Sterne et al., 2016**). The most significant heterogeneity is revealed in the measurement of outcome, while the overall assessment indicated a moderate risk of bias across studies after requesting unreported results and data transformation. Based on the complexity of the type of measures involved in phase-amplitude coupling analysis, we believe that this degree of risk of bias is acceptable. Specific evaluations for each study were reported in Appendix 1.

of 17 overnight and 6 nap studies, in which a total of 19 tasks for declarative memory and 6 tasks for procedural memory were measured. In addition, we assessed the risk of bias for each study before conducting data analysis (see **Figure 3**).

## Coupling phase

We first assessed the association between the preferred phase of SO-SP coupling and memory retention following sleep. 23 studies ($k$ = 90) were included in the Bayesian hierarchical model. We transformed the circular-linear correlation to standardized coefficients (see Methods: Standardized circular-linear correlation coefficient). Forest and regression plots for overall and moderation models are reported in **Figure 4**. In addition, the results of hypothesis tests for the overall and moderator models are reported in **Table 1** using the Bayes factor and posterior probability. Consistent with the funnel plot (**Appendix 4—figure 1A**), neither Egger's regression ($p$ = 0.59) nor rank correlation test ($p$ = 0.52) found the existence of publication bias.

### Overall model

The analysis of a random effect model on the overall phase-memory association revealed that, without considering moderating factors, very strong and consistent evidence supports a small-sized association between the preferred coupling phase and memory consolidation across studies, $r_{z,pooled}$ = 0.07 [0.01, 0.13], $BF_{10}$ = 58.35, probability = 0.98. This implies that the likelihood of $H_1$ is over 58 times greater than $H_0$, covering approximately 98% of posterior samples.

Multilevel model analysis indicates a heterogeneity similar to typical correlational meta-analyses (**Van Erp et al., 2017**) ($M_g$ = 0.13), including a between-study heterogeneity of $g$ = 0.07 [0.00, 0.16], as well as a within-study heterogeneity of $g$ = 0.04 [0.00, 0.11]. Focal analysis showed no difference from the original overall model, with an effect size $r_{z,pooled}$ = 0.07 [−0.06, 0.19]. Sensitivity analysis regarding prior robustness also revealed similar results (see **Appendix 3—table 1**).

### Moderator and sensitivity models

Sufficient evidence shows that moderators including memory tasks, age, spindle types, and PSG channels provide additional predictive power with strong favor for each hypothesis (all contain $BF_{10} \leq 0.1$ or $\geq 10$). **Figure 4C** indicates that most mixed conditions have a considerably higher uncertainty compared to other normal conditions.

### Memory task

Contrary to the assumption, there is no evidence to support a difference of phase-memory association between motor memory ($k$ = 36) and verbal memory retention ($k$ = 35, $r_z$ = Δ0.01, $BF_{10}$ = 1.30 , probability = 0.57), as well as between motor and emotional memory retention ($k$ = 12, $r_z$ = Δ0.01, $BF_{10}$ = 1.09, probability = 0.52). However, strong evidence supports that spatial tasks have a considerably lower phase-memory association compared to motor tasks ($k$ = 7, $r_z$ = Δ0.19, $BF_{10}$ = 0.07, probability = 0.06). The task model has a weight of 0.29, which indicates that the moderator effect of memory task is relatively weak and the task model performed worse than the overall model.

### Participant age

As shown in **Figure 4B**, extremely strong evidence supports that with the increase of age, the slope of correlation exhibits a strongly decreasing trend ($r_{z\beta} = \Delta - 0.006$ [-0.010, -0.001], $BF_{10}$ = 160.94, probability = 0.99). Each ten-year increase in age is associated with a decrease in effect size of 0.06, which is consistent with our hypothesis that precise SO-SP coupling becomes less predictive of memory retention with the increase of age. The moderator effect of age becomes less pronounced during development after excluding the older adult data that represent aging effects, $r_{z\beta} = \Delta - 0.005$ [-0.013, 0.004], $BF_{10}$ = 5.51, showing a moderate effect. Accounting for the factor of age group significantly increased the predictive power compared to the intercept-only model, given a stacked weight of 1.00.

### Spindle frequency

Moderation model with SP frequency range highlights a stronger association between memory retention and SO-fast SP coupling phase ($k$ = 43) rather than slow SPs ($k$ = 28, $r_z = \Delta - 0.07$, $BF_{10}$ = 11.39, probability = 0.92), consistent with the hypothesis based on studies comparing the role of different

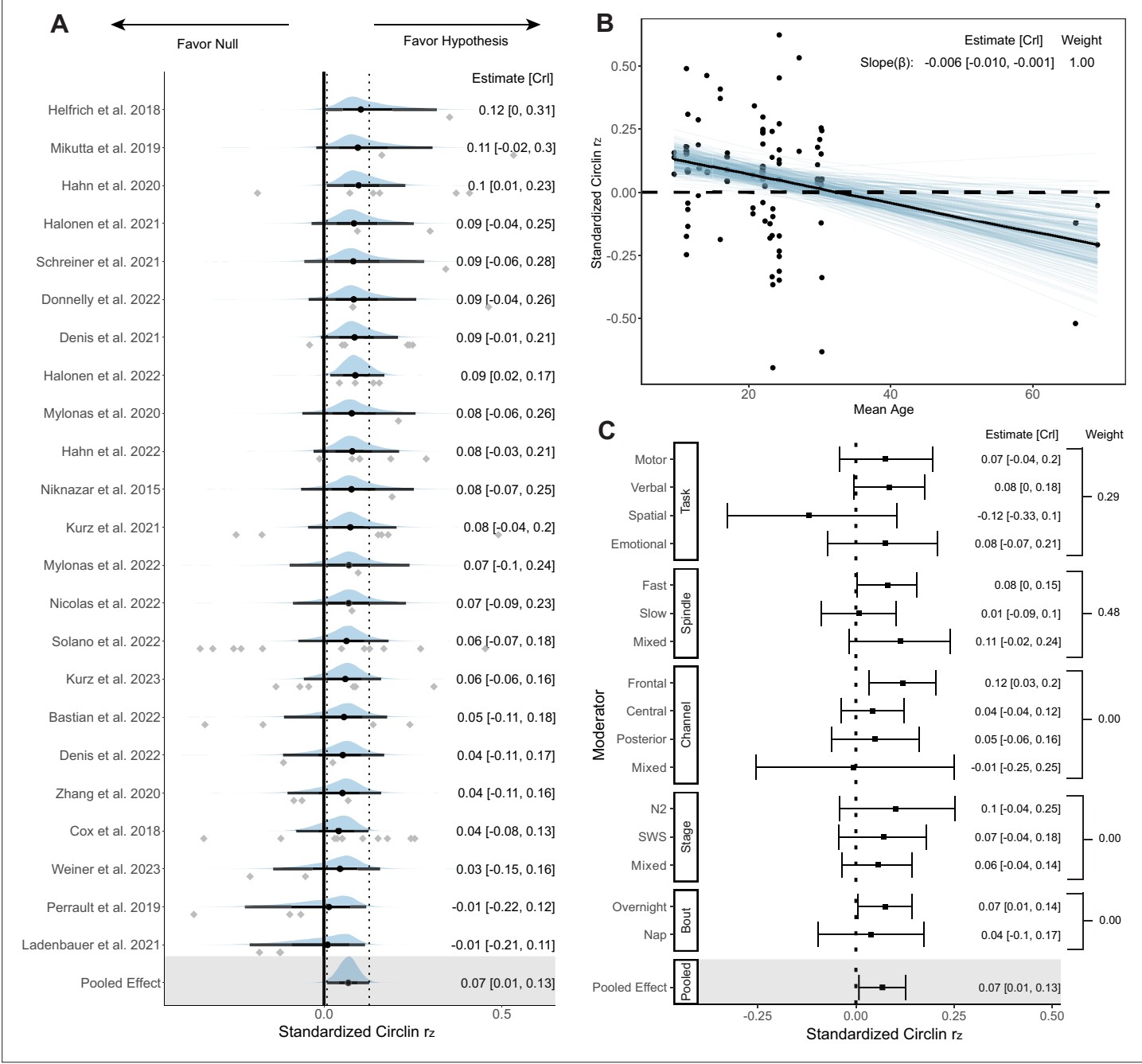

**Figure 4.** Forest and regression plots for the association between SO-SP coupling phase and memory retention. (**A**) Overall model forest plot at study-level. Dashed lines indicate the 95% credible interval (CrI) of the pooled effect size. The black point and error bar for each study show the adjusted estimation of effect size and 95% CrI combining data and prior information. The gray dots under each distribution show raw effect sizes of each study. Effect size-level plots can be found in **Supplementary file 3**. (**B**) Meta regression plot with age as moderator. Blue lines represent 200 overplotted spaghetti fit lines to visualize predictions. (**C**) Moderator-level forest plot. Each box represents a type of moderator. Mixed effect sizes with mixed conditions from different factor levels listed above. *Weight* Stacked weight of each moderation model in the paired model performance comparison between the moderator and overall (intercept-only) model. The stacked weight of the overall model in each pair of comparisons can be calculated as 1 - weight of the moderation model.

The online version of this article includes the following source data for figure 4:

**Source data 1.** Subtable of effect size-level metadata included in the coupling phase–memory analysis.

**Table 1.** Result of directional hypothesis tests for each pair of factor levels (conditions) in overall and each moderation model of the coupling phase-memory association.

| Moderator | Overall | Memory Task | | | Age | Spindle | PSG Channel | | | Stage | Bout |
|---|---|---|---|---|---|---|---|---|---|---|---|
| Condition | $H_1$ | Verbal | Emotional | Spatial | Younger | Fast | Frontal | Frontal | Central | N2 | Night |
| Control | $H_0$ | Motor | Motor | Motor | Older | Slow | Central | Posterior | Posterior | SWS | Nap |
| $BF_{10}$ | 58.35 | 1.30 | 1.09 | 0.07 | 160.94 | 11.39 | 13.58 | 6.13 | 0.86 | 1.74 | 2.23 |
| Probability | 0.98 | 0.57 | 0.52 | 0.06 | 0.99 | 0.92 | 0.93 | 0.86 | 0.46 | 0.63 | 0.69 |

*Condition* conditions hypothesized to be associated with stronger phase-memory association than other factor levels; *Control* Variables hypothesized to be associated with weaker phase-memory association; $BF_{10}$ Bayes factor in favor of $H_1$ over $H_0$; *N2* nREM2 stage; *SWS* slow-wave sleep.

SP types on the memory retention (***Bastian et al., 2022***; ***Mölle and Born, 2011b***). The SP model performed as well as the overall model, weight = 0.48.

### PSG channel

The strongest pooled correlation was observed in the frontal electrode clusters ($k = 29$), compared to the central ($k = 36$, $r_z = \Delta - 0.08$, $BF_{10} = 13.58$, probability = 0.93), and posterior channels ($k = 22$, $r_z = \Delta - 0.07$, $BF_{10} = 6.13$, probability = 0.86), which represents a moderate-to-strong moderation effect. The frontal area has the largest phase-memory association, $r_z = 0.12$ [0.03, 0.21]. Effect sizes in posterior areas do not differ from central areas ($r_z = \Delta - 0.01$, $BF_{10} = 0.86$, probability = 0.46).

### Sleep stage and bout

The sleep stage appeared to be a weak predictor with a weight of 0.00, in which the nREM2 stage ($k = 18$) could not predict a higher phase-memory association than SWS ($k = 30$, $r_z = \Delta - 0.03$, $BF_{10} = 1.74$, probability = 0.63). The sleep bout also revealed no difference in phase-memory association between overnight sleep ($k = 73$) and nap condition ($k = 17$, $r_z = \Delta - 0.03$, $BF_{10} = 2.23$, probability = 0.69), with no predictive role (weight = 0.00) compared to the overall model.

### Spatiotemporal analysis of coupling phase

By aggregating individual-level data across studies, we found that phase of SO-fast SP coupling has a significant quadratic association with memory retention in frontal regions (see ***Figure 5A***), $r = 0.26$, $r_z = 0.20$, $p < 0.01$. However, this association is not significant in either central ($r = 0.11$, $r_z = 0.05$, $p = 0.22$) or posterior regions ($r = 0.08$, $r_z = 0.01$, $p = 0.47$).

Another important finding is a considerable shift of the preferred coupling phase from frontal to posterior regions, similar to ***Kurz et al., 2021***. After taking into account the repeated measurement, the frontal area has a preferred phase around 0.13 rad [0.00, 0.27], which is the closest to the peak of SO (0 rad). The coupling consistently happened earlier when moving towards further dorsally, reflected by the phase in central (−0.27 rad [−0.37, −0.18]) and posterior area (−0.41 rad [−0.54,−0.27]).

Extremely strong evidence supports a considerable difference of the preferred coupling phase between frontal and central areas, $\Delta = -0.40$ rad, $BF_{10} = +\infty$, probability = 1.00; between frontal and posterior areas, $\Delta = -0.54$ rad, $BF_{10} = +\infty$ (An 'infinite' $BF_{10}$ value indicates that all posterior samples are overwhelmingly compatible with $H_1$, given the data and priors. It reflects a reporting convention where $BF_{10}$ becomes unbounded as the likelihood under $H_0$ approaches zero.), probability = 1.00; and between central and posterior areas, $\Delta = -0.14$ rad, $BF_{10} = 34.4$, probability = 0.97. Differences can be observed from the shift of the posterior distribution in ***Figure 5B*** and the location of data clusters in ***Figure 5C***. Moreover, the consistency of coupling phase across subjects also decreases from frontal ($z = 0.69$, $p < 0.05$, Rayleigh test) to posterior areas ($z = 0.59$, $p < 0.05$, Rayleigh test). The differences in coupling phase can explain part of the discrepancy of the timing of coupling occurrence among previous studies (***Bastian et al., 2022***; ***Hahn et al., 2020***; ***Muehlroth et al., 2019***; ***Joechner et al., 2023***).

In summary, we observed that SO-fast SP coupling in the frontal region occurred at the latest phase observed across all regions, characterized by the highest precision and strength of phase-locking. We

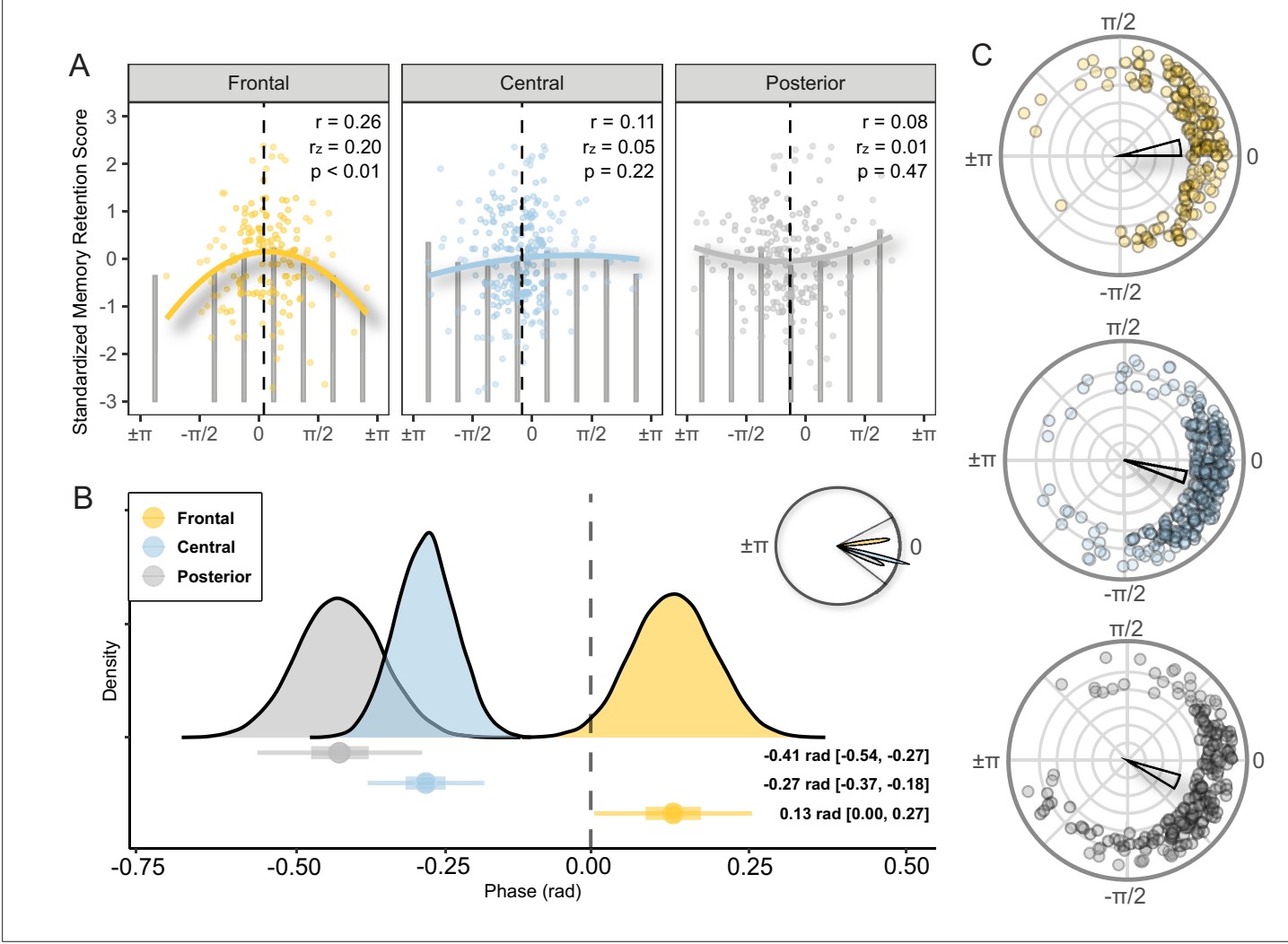

**Figure 5.** Preferred slow oscillation-fast spindle coupling phase and its association with memory retention. (**A**) Quadratic regression of the phase-memory association under different regions of PSG channels aggregated from studies included in the meta-analysis. *0* peak of SO upstate; ±*π* trough of SO downstate; *r* circular-linear correlation coefficient; *r*$_z$ standardized circular-linear correlation coefficient. Bars represent the mean memory retention scores per *π*/4 radian (45°). The dashed vertical line represents the mean preferred phase across studies. The colored quadratic fit line represents the direction of the relationship. The direction of their relationship gradually flips as the PSG channel moves from the front to the posterior area. Only under the frontal and central channels do fit lines display a quadratic relationship, with a peak of memory score near the up-state peak of SOs. In posterior channels, in contrast, the relationship is convex, although it is not significant. Non-significant quadratic regressions between SO-slow SP coupling and memory are reported in Appendix 6. (**B**) Posterior distributions of mean preferred phases from the Bayesian circular mixed-effect model. The circular posterior distribution is shown in the top-right corner, and the area between two black lines is projected on a linear scale in the main graph. The vertical line reflects the up-state peak of SOs. Points and error bars denote the mean and 95% credible intervals of phases detected from each channel cluster. Phase values are reported as radians. (**C**) Circular plot of the preferred coupling phase. From top to bottom, frontal, central, and posterior. The direction of each colored dot represents the preferred coupling phase of each subject recorded from PSG channels in each cluster. The direction of the mean resultant vector indicates the mean preferred coupling phase across subjects, the width indicates the 95% credible interval of the mean coupling phase, the length from 0 (center) to 1 (circumference) indicates the consistency of coupling phase across subjects.

also found the strongest phase-memory association in frontal regions following a typical quadratic relationship.

## Spindle amplitude

We next assessed the association between the amplitude/power of SP and memory retention during the learning night. 18 original studies (*k* = 78) were included in the Bayesian hierarchical model. The mean amplitude of fast SPs (in μV) is 33.99 [26.44,41.55], while the mean amplitude of slow SPs is

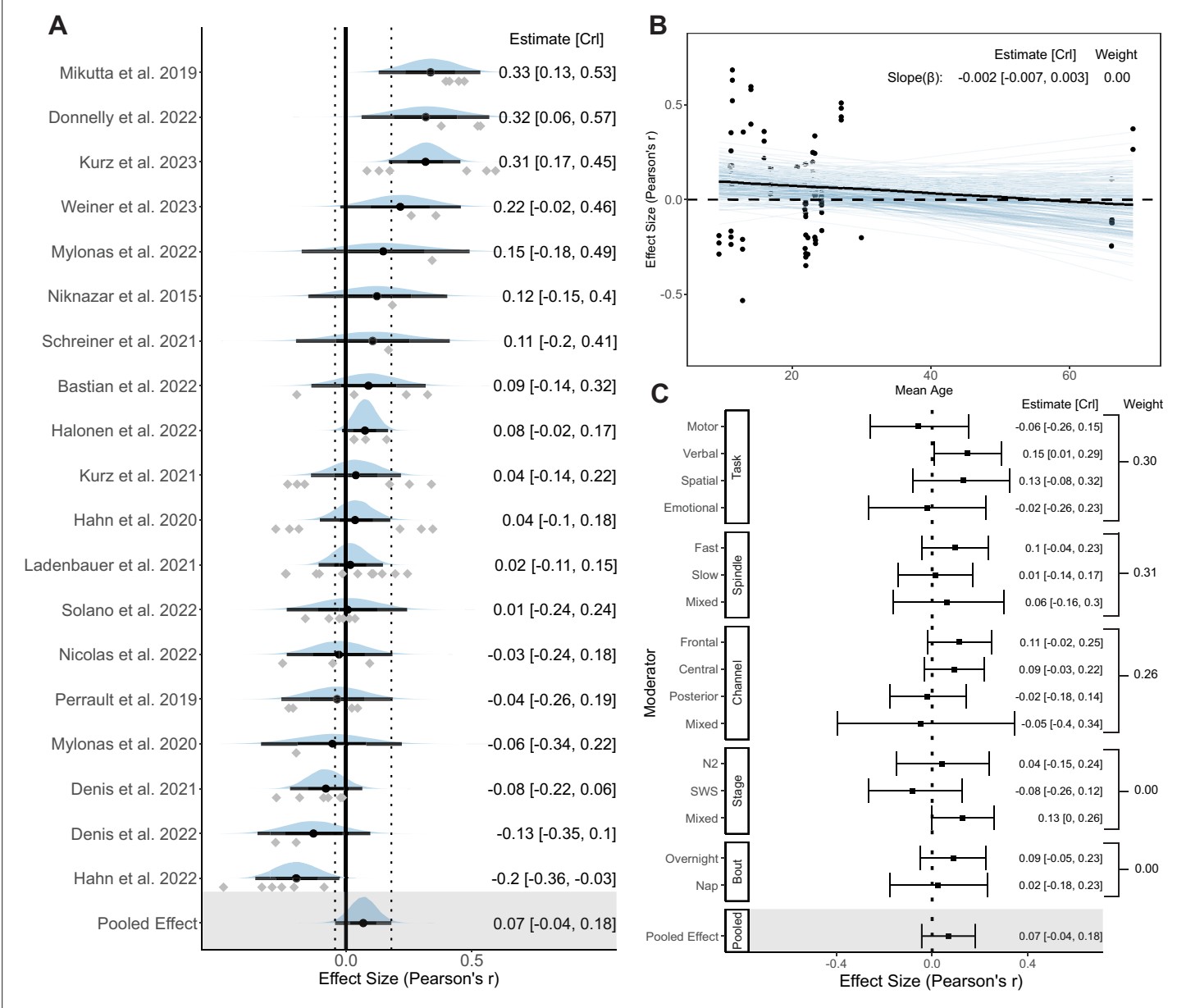

**Figure 6.** Forest and regression plots for the association between SP amplitude and memory retention. (**A**) Overall model forest plot at study-level. The solid vertical line represents the mean Pearson correlation coefficient under the null hypothesis. Dashed lines indicate the 95% credible interval (CrI) of the pooled effect size. The black point and error bar for each study show the adjusted estimation of effect size and 95% CrI combining data and prior information. The gray dots under each distribution show raw effect sizes of each study. Effect size-level plots can be found in *Supplementary file 3*. (**B**) Meta regression plot with age as moderator. Blue lines represent 200 overplotted spaghetti fit lines to visualize predictions. (**C**) Moderator-level forest plot. Each box represents a type of moderator. Mixed effect sizes with mixed conditions from different factor levels listed above. *Weight* Stacked weight of each moderation model in the paired model performance comparison between the moderator and overall (intercept-only) model. The stacked weight of the overall model in each pair of comparisons can be calculated as 1 weight of the moderation model.

The online version of this article includes the following source data for figure 6:

**Source data 1.** Subtable of effect size-level metadata included in the spindle amplitude memory analysis.

39.41 [33.61, 45.22]. Forest and regression plots for the overall and moderation models are reported in *Figure 6*. In addition, the results of hypothesis tests for the overall and moderator models are reported in *Table 2*. The publication bias of the amplitude-memory association studies is unclear but potentially provides evidence of asymmetry. Egger's regression test (*p* = 0.02) indicated potential

**Table 2.** Result of directional hypothesis tests for each pair of factor levels (conditions) in overall and each moderation model of the SP amplitude-memory association.

| Moderator | Overall | Memory Task | | | Age | Spindle | PSG Channel | | | Stage | Bout |
|---|---|---|---|---|---|---|---|---|---|---|---|
| Condition | $H_1$ | Verbal | Emotional | Spatial | Younger | Fast | Frontal | Frontal | Central | N2 | Night |
| Control | $H_0$ | Motor | Motor | Motor | Older | Slow | Central | Posterior | Posterior | SWS | Nap |
| $BF_{10}$ | 8.28 | 22.19 | 1.50 | 9.30 | 3.70 | 12.21 | 1.88 | 24.67 | 13.65 | 5.37 | 2.54 |
| Probability | 0.89 | 0.96 | 0.60 | 0.90 | 0.79 | 0.92 | 0.65 | 0.96 | 0.93 | 0.84 | 0.72 |

*Condition* conditions hypothesized to be associated with stronger amplitude-memory association than other factor levels; *Control* Variables hypothesized to be associated with weaker amplitude-memory association; $BF_{10}$ Bayes factor in favor of $H_1$ over $H_0$; *N2* nREM2 stage; *SWS* slow-wave sleep.

publication bias, whereas the rank correlation test ($p = 0.11$) and funnel plot (**Appendix 4—figure 1C**) did not suggest significant bias.

## Overall model

We observed that without accounting for moderation effects, there was moderate evidence supporting a positive association between SP peak-to-trough amplitude and memory consolidation, $r_{pooled} = 0.07$ [$-0.04$, $0.18$], $BF_{10} = 8.28$, probability = 0.89. The likelihood of our hypothesis being true is about 8 times greater than the null, covering approximately 89% of posterior samples. However, the pooled posterior distribution spans a wide range due to a large standard error, 95% credible intervals of most studies, and the pooled effect size including 0. Three-level analysis has found a relatively high between-study heterogeneity of $g = 0.20$ [$0.12$, $0.32$], and a normal level of within-study heterogeneity of $g = 0.04$ [$0.00$, $0.12$]. The focal model indicated a slightly larger effect size compared to the overall model, with a pooled effect size $r_{pooled} = 0.12$ [$-0.03$, $0.29$]. Prior sensitivity analysis did not reveal different patterns of the association (see **Appendix 3—table 2**).

## Moderator and sensitivity models

Although evidence shows that memory type, spindle type, and PSG channel location modulates the magnitude of correlation between SP amplitude and memory ($BF_{10} \leq 0.1$ or $\geq 10$), it is worth noting that all moderation models did not provide enough additional information to increase the performance of model prediction relative to the overall model (all weights $\leq 0.31$).

### Memory task

The result of memory task moderation model shows that overnight retention of declarative tasks, including verbal tasks ($k = 34$, $r = \Delta0.21$, $BF_{10} = 22.19$, probability = 0.96); emotional tasks ($k = 12$, $r = \Delta0.04$, $BF_{10} = 1.50$, probability = 0.06); and spatial tasks ($k = 14$, $r = \Delta0.19$, $BF_{10} = 9.30$, probability= 0.90), have a higher association with the SP amplitude compared to the motor memory retention ($k = 36$). It is worth noting that two types of hippocampus-dependent memory, including verbal and spatial memory, both have larger estimates compared to the non-hippocampus-dependent memory. However, only the comparison between verbal and motor tasks provided strong favor for our hypothesis. Strong evidence supports a positive association between the SP amplitude and verbal declarative memory retention, $r = 0.15$ [$0.01$, $0.29$].

### Spindle frequency

The model results using the SP type as the moderator indicate a stronger positive relationship between fast SP amplitude ($k = 36$) and memory retention compared to slow SP amplitude ($k = 20$, $r = \Delta - 0.09$, $BF_{10} = 12.21$, probability = 0.92). A higher fast SP amplitude tends to be associated with a higher memory retention ability, $r = 0.1$ [$-0.04$, $0.23$].

### PSG channel

Strong evidence supports differences of pooled correlation between the frontal ($k = 29$) and posterior region ($k = 16$, $r = \Delta - 0.13$, $BF_{10} = 24.67$, probability = 0.96), as well as between the central

($k = 30$) and posterior region ($r = \Delta - 0.11$, $BF_{10} = 13.65$, probability $= 0.93$), which is consistent with our assumptions. Frontal regions have the strongest positive amplitude-memory association, $r = 0.11$ [$-0.02$, $0.25$]. In contrast, we did not observe a notable difference in amplitude-memory association between frontal and central regions ($r = \Delta - 0.02$, $BF_{10} = 1.88$, probability $= 0.65$).

### Other moderates

There is moderate evidence supporting a difference in the amplitude-memory association between N2 and SWS stage ($r = \Delta - 0.12$, $BF_{10} = 5.37$, probability $= 0.84$), but no reliable difference between overnight and nap conditions ($r = \Delta - 0.07$, $BF_{10} = 2.54$, probability $= 0.72$). Also, the effect of age on the prediction of amplitude-memory association is highly limited ($r_\beta = \Delta - 0.002 [-0.007, 0.003]$, $BF_{10} = 3.70$, probability $= 0.79$).

## Coupling strength

Next, we measured the association between coupling strength and memory retention. 21 original studies ($k = 86$) were included in the Bayesian hierarchical model. The mean SO-fast SP coupling strength measured by mean vector length is 0.33 [0.27, 0.39], while the mean SO-slow SP coupling strength is 0.23 [0.19, 0.27]. Forest and regression plots for the overall and moderation models are reported in *Figure 7*. In addition, the results of hypothesis tests for the overall and moderation models are reported in *Table 3*. Neither Egger's regression ($p = 0.53$) nor rank correlation test ($p = 0.67$) showed evidence of potential publication biases.

### Overall model

Consistent with results of the overall phase-memory association, extremely strong evidence supports a positive coupling strength-memory association in the intercept-only model ($r_{\text{pooled}} = 0.08 [0.02, 0.15]$, $BF_{10} = 111.04$, probability $= 0.99$), which strongly favors our hypothesis. The likelihood of data under our hypothesis is over 110 times greater than the negative direction, covering above 99% of posterior samples.

Similar to typical correlational meta-analyses, the overall model consists of a between-study heterogeneity of $g = 0.08 [0.01, 0.17]$, and a within-study heterogeneity of $g = 0.05 [0.00, 0.14]$. The result of focal analysis showed no difference from the overall model but has a wider credible interval due to a smaller number of effect sizes, $r_{\text{pooled}} = 0.09 [-0.02, 0.20]$. Sensitivity analysis regarding prior robustness revealed consistent results with the overall model (see *Appendix 3—table 3*).

### Moderator and sensitivity models

Surprisingly, there is not enough evidence supporting a difference between each pair of factor levels for almost all moderators (all $0.1 < BF_{10} < 10$, except emotional versus motor tasks), and there was no single moderator model that had better performance than the intercept-only model (all weights $< 0.5$). This result represents a relatively consistent strength-memory association, regardless of the impact of moderators.

**Table 3.** Result of directional hypothesis tests for each pair of factor levels (conditions) in overall and each moderation model of the coupling strength-memory association.

| Moderator | Overall | Memory Task | | | Age | Spindle | PSG Channel | | | Stage | Bout |
|---|---|---|---|---|---|---|---|---|---|---|---|
| Condition | $H_1$ | Verbal | Emotional | Spatial | Younger | Fast | Frontal | Frontal | Central | N2 | Night |
| Control | $H_0$ | Motor | Motor | Motor | Older | Slow | Central | Posterior | Posterior | SWS | Nap |
| $BF_{10}$ | 111.04 | 2.77 | 10.62 | 1.10 | 9.97 | 1.91 | 2.81 | 1.57 | 0.70 | 0.33 | 2.66 |
| Probability | 0.99 | 0.73 | 0.91 | 0.52 | 0.91 | 0.66 | 0.74 | 0.61 | 0.41 | 0.25 | 0.73 |

*Condition* conditions hypothesized to be associated with stronger strength-memory association than other factor levels; *Control* Variables hypothesized to be associated with weaker strength-memory association; $BF_{10}$ Bayes factor in favor of $H_1$ over $H_0$; *N2* nREM2 stage; *SWS* slow-wave sleep.

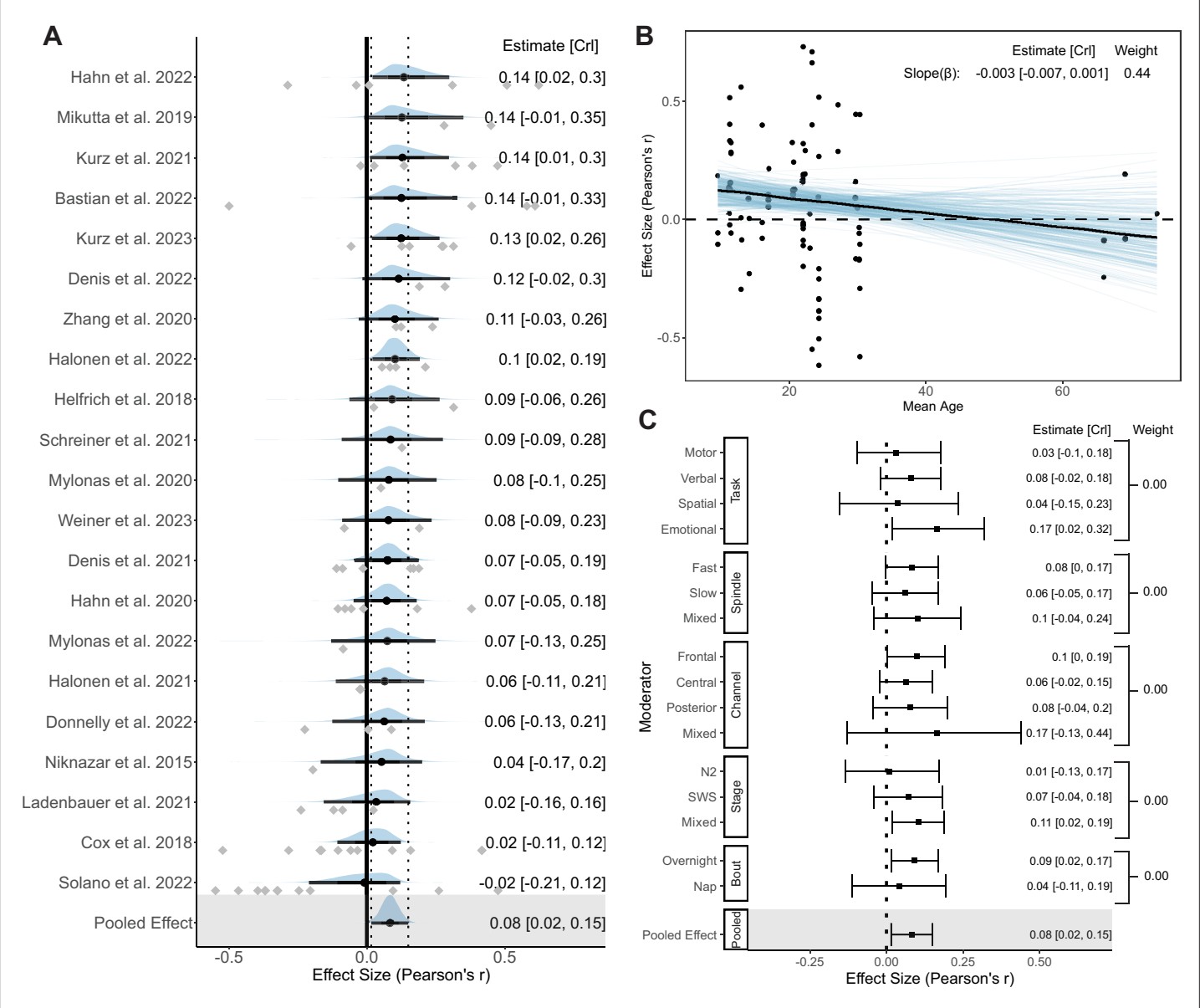

**Figure 7.** Forest and regression plots for the association between coupling strength and memory retention. (**A**) Overall model forest plot at study-level. The solid vertical line represents the mean Pearson correlation coefficient under the null hypothesis. Dashed lines indicate the 95% credible interval (CrI) of the pooled effect size. The black point and error bar for each study show the adjusted estimation of effect size and 95% CrI combining data and prior information. The gray dots under each distribution show raw effect sizes of each study. Effect size-level plots can be found in *Supplementary file 3*. (**B**) Meta regression plot with age as moderator. Blue lines represent 200 overplotted spaghetti fit lines to visualize predictions. (**C**) Moderator-level forest plot. Each box represents a type of moderator. Mixed effect sizes with mixed conditions from different factor levels listed above. *Weight* Stacked weight of each moderation model in the paired model performance comparison between the moderator and overall (intercept-only) model. The stacked weight of the overall model in each pair of comparisons can be calculated as 1 weight of the moderation model.

The online version of this article includes the following source data for figure 7:

**Source data 1.** Subtable of effect size-level metadata included in the coupling strength–memory analysis.

### Memory task

With the exception of the emotional task condition ($k = 12$) which predicts a larger strength-memory association than the motor task condition ($k = 32$, $r = \Delta - 0.20$, $BF_{10} = 10.62$, probability = 0.91), the type of memory task only had a weak impact on the association (all other $0.33 < BF_{10} < 3$). In addition,

given the emotional task condition has a relatively small number of effect sizes, the evidence of group differences also needs to be interpreted with caution.

### Age model

The strongest predictive power among moderation models has been found with participants' age, and performs similarly in the trade-off between complexity and power with the intercept-only model (weight = 0.44). As the age increases, the slope of the correlation exhibits a decreasing trend that is nearly in strong favor of our hypothesis ($r_\beta = \Delta - 0.003\,[-0.007, 0.001]$, $BF_{10} = 9.97$, probability = 0.91). The strength-memory association becomes weaker with the increase of age. The effect of age remains moderate after removing older adults from the analysis, $r_\beta = \Delta - 0.005\,[-0.015, 0.008]$, $BF_{10} = 4.05$.

### Other moderators

There was no strong evidence of a difference in strength-memory association between the fast ($k = 41$) and slow SPs ($k = 26$, $r = \Delta - 0.02$, $BF_{10} = 1.91$, probability = 0.66). Also, there is no evidence for a group difference among any pair of conditions of PSG channels, sleep stages, or sleep bout type (all $0.33 < BF_{10} < 3$). However, consistent with the results for the coupling phase and SP amplitude, the largest positive strength-memory association moderated by PSG channels was still found in frontal channels ($r = 0.10\,[0.00, 0.19]$).

## Coupling percentage

The last part of our analysis focused on the association between the percentage of SPs coupled with SOs and memory consolidation. 11 studies ($k = 43$) were included in the Bayesian hierarchical model. The mean SO-fast SP coupling percentage (%) is 21.17 [15.95, 26.39], while the SO-slow SP coupling percentage (%) is 24.07 [17.59, 30.56]. Forest and regression plots for the overall and moderator models are reported in *Figure 8*. In addition, the results of hypothesis tests for the overall and moderator models are reported in *Table 4*. Consistent with the funnel plot (*Appendix 4—figure 1E*), both Egger's regression ($p = 0.78$) and rank correlation test ($p = 0.87$) showed no publication bias.

### Overall model

Compared to all other coupling measures, we observed a weak association between coupling percentage and memory consolidation ($r_{\text{pooled}} = -0.03\,[-0.15, 0.07]$, $BF_{10} = 0.38$, probability = 0.28). The overall model consists of a moderate between-study heterogeneity of $g = 0.11\,[0.01, 0.25]$, and a within-study heterogeneity of $g = 0.05\,[0.00, 0.14]$. 95% credible intervals of all studies and the pooled effect size include 0. Due to the limited number of effect sizes, we did not perform a focal analysis. Prior sensitivity analysis revealed consistent results with the overall model (see *Appendix 3—table 4*).

### Moderator and sensitivity models

Few moderators had an influential impact on the percentage-memory association (All $0.1 < BF_{10} < 10$, except verbal versus motor tasks, all weights < 0.40), which is not surprising given the weak association in the overall model. For memory types, we observed a moderate trend of declarative memory retention towards a negative association with coupling percentage, in contrast to the positive trend of percentage-motor memory association. Therefore, the association measured by motor memory tasks

**Table 4.** Results of directional hypothesis tests for each pair of factor levels (conditions) in overall and each moderation model of the coupling percentage–memory association.

| Moderator | Overall | Memory Task | | | Age | Spindle | PSG Channel | | | Stage | Bout |
|---|---|---|---|---|---|---|---|---|---|---|---|
| Condition | $H_1$ | Verbal | Emotional | Spatial | Younger | Fast | Frontal | Frontal | Central | N2 | Night |
| Control | $H_0$ | Motor | Motor | Motor | Older | Slow | Central | Posterior | Posterior | SWS | Nap |
| $BF_{10}$ | 0.38 | 0.05 | 0.17 | 0.22 | 2.16 | 0.28 | 5.15 | 3.38 | 1.24 | 2.02 | 0.44 |
| Probability | 0.28 | 0.05 | 0.14 | 0.18 | 0.68 | 0.22 | 0.84 | 0.77 | 0.55 | 0.67 | 0.31 |

*Condition* conditions hypothesized to be associated with stronger percentage-memory association than other factor levels; *Control* Variables hypothesized to be associated with weaker percentage memory association; $BF_{10}$ Bayes factor in favor of $H_1$ over $H_0$; *N2* nREM2 stage; *SWS* slow-wave sleep.

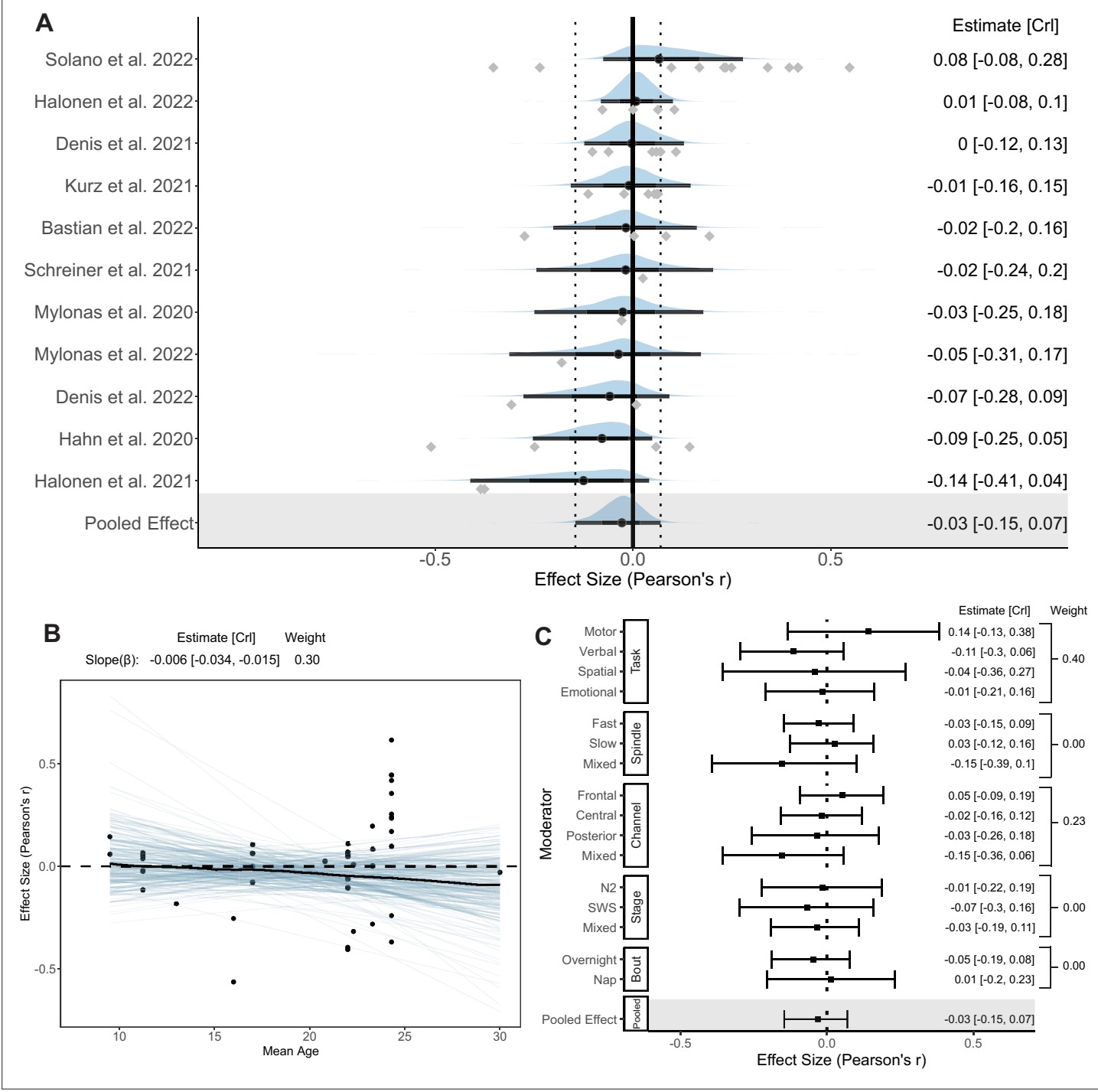

**Figure 8.** Forest and regression plots for the association between coupling percentage and memory retention. (**A**) Overall model forest plot at study-level. The solid vertical line represents the mean Pearson correlation coefficient under the null hypothesis. Dashed lines indicate the 95% credible interval (CrI) of the pooled effect size. The black point and error bar for each study show the adjusted estimation of effect size and 95% CrI combining data and prior information. The gray dots under each distribution show raw effect sizes of each study. Effect size-level plots can be found in *Supplementary file 3*. (**B**) Meta regression plot with age as moderator. Blue lines represent 200 overplotted spaghetti fit lines to visualize predictions. (**C**) Moderator-level forest plot. Each box represents a type of moderator. Mixed effect sizes with mixed conditions from different factor levels listed above. *Weight* Stacked weight of each moderation model in the paired model performance comparison between the moderator and overall (intercept-only) model. The stacked weight of the overall model in each pair of comparisons can be calculated as 1 weight of the moderation model.

The online version of this article includes the following source data for figure 8:

**Source data 1.** Subtable of effect size-level metadata included in the coupling percentage–memory analysis.

**Table 5.** Descriptive table of the meta-analysis result of measures of SO-SP coupling characteristics.

| Measure | n | k | Pooled Effect Size | BF$_{10}$ | Influential Moderators |
|---|---|---|---|---|---|
| Coupling Phase | 23 | 90 | 0.07 [0.01, 0.13] | 58.35 | Memory Type, Age, Channel, Spindle |
| Spindle Amplitude | 18 | 78 | 0.07 [−0.04, 0.18] | 8.28 | Memory Type, Age, Channel, Spindle |
| Coupling Strength | 22 | 86 | 0.08 [0.02, 0.15] | 111.04 | Age |
| Coupling Percentage | 11 | 43 | −0.03 [−0.15, 0.07] | 0.38 | None |

*n* number of studies included; *k* number of effect sizes included; *BF$_{10}$* Bayes factor in favor of $H_1$ over $H_0$ (see **Table 8** for the interpretation).

($k = 13$) is considerably higher than verbal tasks, $k = 13$, $r = \Delta - 0.25$, $BF_{10} = 0.05$, probability = 0.05. All other moderators did not trend toward a specific direction, and all 95% credible intervals include 0.

## Discussion

As the first meta-analysis focusing on the coupling between slow oscillation and spindle events, our results, combining 297 effect sizes, provide reliable evidence for the involvement of thalamocortical SO-SP coupling in memory consolidation. In particular, the precision and strength of coupling, represented by the preferred phase and coupling strength, showed significant effect sizes in their correlation with memory retention performance. Moderators, including age, memory type, cortical area, and SP frequency, modulated the magnitude of these associations. The conclusive results are summarized in **Table 5**.

### Precision and strength of SO-SP coupling as strong predictors of memory consolidation

In previous meta-analyses focusing on spindle events, the amplitude and power of SPs have been considered the most predictive measures for memory consolidation (**Kumral et al., 2023**) and cognitive abilities (**Ujma, 2021**). However, our findings indicate that the SP amplitude-memory association is subject to high variability. Only the association between fast SP amplitude and hippocampal-dependent memory consolidation is supported by strong evidence. In contrast, the precision and strength of coupling between the fast SP peak amplitude and the up-state peak of SOs are more crucial indicators for predicting memory retention performance. Our result confirmed the importance of cross-frequency coupling in the hierarchical temporal nesting within the hippocampus-thalamus-cortex information transmission loop, proposed in the active system consolidation theory (**Klinzing et al., 2019**; **Staresina et al., 2015**; **Born and Wilhelm, 2012**; **Klinzing et al., 2016**). Together with other studies included in the review (**Hahn et al., 2020**; **Helfrich et al., 2018**; **Niknazar et al., 2015**; **Muehlroth et al., 2019**; **Mölle and Born, 2011b**), our results suggest a crucial role of coupling but did not support the role of spindle events alone in memory consolidation.

Moreover, we found that the phase and strength of coupling between fast SPs and SOs retain strong predictive ability for memory retention performance in most sub-groups of the moderator analysis. This result confirms the robustness of the connection between coupled SO-fast SP phase and memory consolidation as a general physiological mechanism. Similar results were not observed with slow SPs. Since we did not find publication bias in both analyses regarding coupling phase and strength, it reduces the likelihood of overestimating true effect sizes. Given we have identified predictive powers for memory retention in both coupling phase and strength, and **Weiner et al., 2024** reported a synergistic interaction between phase and strength, future studies are necessary to further investigate the relationship between these phase-amplitude level measures.

### The modulation role of cortical area and oscillation frequency

Most of the studies included in the meta-analysis supported the significance of the association between the coupling phase and memory consolidation. However, conflicting conclusions have been reported regarding the direction of the phase. In terms of the preferred coupling phase, we find two predominant views, with a subset of studies reporting that coupling events are concentrated before the up-state peak ($-\pi/2$ to 0) of SOs (**Cox et al., 2018**; **Kurz et al., 2021**; **Niknazar et al., 2015**;

*Perrault et al., 2019*; *Ladenbauer et al., 2021*; *Schreiner et al., 2021*). Other authors found that the preferred phase is when fast SPs are precisely coupled with the up-state peak (0) (*Hahn et al., 2020*; *Halonen et al., 2021*; *Helfrich et al., 2018*; *Mikutta et al., 2019*; *Mylonas et al., 2020*; *Mylonas et al., 2022*). In either case, however, most of the studies agreed that coupling occurring closer to the up-state peak of SOs could predict better memory retention performance.

In addition to differences in signal processing approaches and age leading to discrepancies in phases reported, we believe that another reason is different PSG electrodes and SP frequencies filtered to detect coupling events. Research found that fast SPs occur more frequently in centroparietal regions, while slow SPs are predominantly in frontal regions (*Mölle et al., 2011a*; *Anderer et al., 2001*; *De Gennaro and Ferrara, 2003*; *Urakami, 2008*). This evidence led some research to apply this conclusion to coupling studies and detecting fast SPs only from centro-parietal electrodes or slow SPs only from frontal electrodes (*Bastian et al., 2022*; *Niknazar et al., 2015*; *Perrault et al., 2019*; *Schreiner et al., 2021*). However, we suspect that this approach restricts researchers from considering the origin and spread of oscillations. The prefrontal cortex has been proposed to be responsible for the generation of posterior-propagating global SOs (*Staresina et al., 2015*; *Massimini et al., 2004*; *Marshall et al., 2006*; *Achermann and Borbély, 1997*), while global SPs, the majority of SP events, propagate in a rotating direction following gradients (*Dickey et al., 2021*; *Muller et al., 2016*; *O'Reilly and Nielsen, 2014*). The significance of this dynamic interaction of oscillations for spatiotemporal coordination remains poorly understood.

SO-SP coupling can be widely detected across the frontoparietal cortex (*Fernandez and Lüthi, 2020*), and the interaction between centroparietal-dominated fast SPs and frontal-dominated SOs, along with their propagations, is proposed to be crucial for long-range transmission of memory-related information from thalamus to neocortex (*Maingret et al., 2016*; *Marshall et al., 2006*; *Helfrich et al., 2019*; *Hahn et al., 2020*; *Baena et al., 2023*). Our spatiotemporal analysis provides extremely strong evidence that the post-peak SO-fast SP coupling phase recorded from frontal regions occurs significantly closer to the up-state peak of SOs and considerably later than the pre-peak phase in centroparietal regions, which supports phase shifts across electrodes found in previous studies (*Cox et al., 2018*; *Denis et al., 2022*; *Denis et al., 2021*), emphasizing the importance of coupling precision but not intensity. Moderator models of all four coupling measures consistently indicate that coupling detected from frontal regions has the largest association with memory consolidation. Moreover, a significant quadratic phase-memory relationship is only observed in frontal regions. In addition, the frontal region has the strongest and most active SOs as its origin site (*Massimini et al., 2004*; *Kurth et al., 2017*; *Malerba et al., 2019*; *Niethard et al., 2018*), which may contribute to the role of frontal coupling. To the best of our knowledge, our study is the first to provide consistent evidence supporting that successful memory consolidation is modulated by spatiotemporal specificity, emphasizing the precise coupling of frontal fast SPs targeting the up-state peak of SOs.

## Further moderation through memory types, aging, and sleep conditions

The predictive ability of coupling for memory consolidation has been studied across various types of memory, including emotional, spatial, verbal, and motor tasks. Surprisingly, our results indicate that the coupling phase has a widespread predictive role for both declarative and procedural memory. In contrast, evidence only supports a correlation between coupling strength and hippocampus-dependent memory. In addition, *Hahn et al., 2022* and *Hahn et al., 2020* provided a region-specific view, claiming significant associations of SO-SP coupling with declarative and procedural memory exist in the frontal lobe and motor cortex, respectively. Therefore, it is worthwhile for future research to study both shared mechanisms and region specificity of coupling across different memory types, including those traditionally considered non-hippocampus-dependent. However, we must exercise caution that the number of effect sizes extracted for emotional and spatial tasks is limited, making it susceptible to the influence of any single study.

One disadvantage in our analysis of age as a moderator is the limited number of studies on children and older adults. The linear meta-regression as a function of age did not capture the moderating pattern of development but only reflected aging through the decreasing trend of the coupling-memory association. For older adults, the meta-regression showed that all four measures approached null hypothesis levels of non-significance, indicating a diminishing effect in predicting memory

consolidation through coupling phase or strength with aging. These results support previous studies on memory consolidation impairment in older adults (*Helfrich et al., 2018*; *Muehlroth et al., 2019*). While our results did not directly reflect the influence of development on the coupling-memory association, evidence from past research suggested that the strength of SO-SP coupling in frontal lobes increases with developmental age, indicating the development of long-term memory networks (*Hahn et al., 2020*). We suggest future studies to further compare SO-SP coupling across different age groups, contrasting the regional distribution of coupling changes during development and aging.

However, evidence did not support the moderation role of stage conditions, including sleep stages and bouts, in the coupling-memory association, which is somewhat surprising. SOs are predominant in SWS (*Dang-Vu et al., 2008*), while sleep SPs are the primary oscillation feature in N2 (*De Gennaro and Ferrara, 2003*). Our included studies showed that the co-occurrence rate of SO-SP coupling is higher during SWS compared to N2. One possible explanation, as suggested by our meta-analysis results, is that the precision and strength of coupling is more predictive of memory consolidation than the coupling percentage. Thus, considering the contribution of different stages to memory consolidation through co-occurrence rates may not be meaningful. Regarding sleep bouts (naps versus overnight sleep), no differences were found in any coupling measures between overnight sleep and naps, consistent with the results of a meta-analysis focusing on spindle-memory association (*Kumral et al., 2023*; *Schmid et al., 2020*). It implies the potential benefits of napping in memory consolidation and the clearance of hippocampal traces for storing new knowledge during the daytime.

## Challenges of current statistical approaches in measuring EEG-behavior associations

One of the crucial factors limiting further interpretation of our results is the variation in statistical methods across studies, including differences in the definition of behavioral and physiological measures, the event detection and analysis methods employed, and limitations in estimating nonlinear relationships. Firstly, we agree with (*McConnell et al., 2021*) that there has been confusion in previous studies, where the term "slow spindle" is inconsistently used to refer to either frontal SPs (~10–14 Hz) or low-frequency slow SPs (~7–11 Hz). Compounding this issue, individual differences in spindle frequency are often overlooked, leading to challenges in reliably distinguishing between slow and fast spindles. Our result reveals that frontal fast SPs (~10–14 Hz) occurring near the up-state peak of frontal-predominant global SOs with lower frequency than the centro-parietal fast SPs (~14–17 Hz), which becomes predominant with development (*Hahn et al., 2019*; *Shinomiya et al., 1999*; *McConnell et al., 2022*; *McConnell et al., 2021*). Some studies have also reported difficulty in separating these two types of spindles (*Hahn et al., 2020*). In contrast, slow SP occurs before the trough of the SO down-state. Our findings support a detailed categorization of SP types proposed in previous studies (*McConnell et al., 2021*; *Bernardi et al., 2018*; *McConnell et al., 2022*; *Siclari et al., 2014*), involving separate measures between pre-peak early-fast SPs, post-peak (or peak) late-fast SPs (see *Figure 5C*), and pre-trough slow SPs (see *Appendix 6—figure 1C*). Alternatively, an effective approach could involve extracting signals from electrodes across different cortical areas for each type of SP for comparison. In addition, the current definition of moderators is also quite vague and conflicted. Some studies conducted analyses across mixed conditions, including using a global frequency range (e.g., 10–16 Hz) for SP detection, as well as reporting only one shared effect size for all conditions.

Moreover, we observed significant between-study differences in the measures for memory retention performance. Specifically, multiple retention measures already exist when solely considering verbal tasks for measuring declarative memory, including (1) the value difference in the number of correctly recalled words (Post-sleep – Pre-sleep number of correct words) (*Weiner et al., 2024*; *Niknazar et al., 2015*; *Perrault et al., 2019*); (2) the value difference in the percentage of correctly recalled words (Post-sleep – Pre-sleep % of correct words) (*Hahn et al., 2020*; *Zhang et al., 2020*; *Ladenbauer et al., 2021*); and (3) the ratio difference in the percentage of correctly recalled words (Post-sleep / Pre-sleep % of correct words) (*Denis et al., 2021*; *Mikutta et al., 2019*; *Schreiner et al., 2021*). Similar situations exist in measures for other memory retention tasks. When we tested the sleep-memory association using these three formulas separately, we found that in some extreme cases, different measures could even alter the direction of the association. Thus, we strongly suggest the consistency in memory retention measures in future studies (*Németh et al., 2024*).

In addition, misinterpretation of phase-memory associations in some studies poses a threat to the validity of results. Most studies use circular-linear correlation to measure this association, but we observed a prevalence of exaggeration when explaining the effect size. Circular-linear correlations lack directionality (*Mardia, 1976*), making it challenging to precisely estimate the improvement or decay of memory consolidation at specific SO phases. It is likely to be influenced by fluctuations in memory scores in any segment of SO phases, so associating it with hypotheses targeting specific phases might lead to incorrect conclusions. An effective solution is to visualize the regression and superimpose quadratic fit lines to assess whether it follows a typical quadratic relationship around the global maximum or minimum. In addition, through simulation studies (see Appendix 5), we observed that circular-linear correlation coefficients exhibit weak robustness and severe deviation from normal distribution for small to medium sample sizes ($n < 100$), commonly encountered limitations in PSG studies.

To suppress the bias introduced by non-linear relationships, we believe there are two available solutions: (1) *Hahn et al., 2020* calculated the absolute distance of the preferred phase of each participant from the upstate peak (0), while *Kurz et al., 2023* and *Weiner et al., 2024* applied this method to transform the phase-memory association into a linear relationship for subsequent testing. Our results provided solid support for using SO up-state peaks as the center of transformation. (2) We developed a method to standardize the circular-linear correlation coefficient by transforming the sampling distribution of correlation to be normally distributed and centered at 0 under null, which takes into account the sample size to eliminate the overestimation of effect sizes and becomes comparable with Pearson's correlation to enhance the comparability across studies (see Methods: Standardized circular-linear correlation coefficient).

In reporting results, despite the tendency to introduce multiple comparison issues due to the presence of multiple time points and electrodes during PSG recording (*Yang et al., 2018*), less than half of the included studies conducted corrections for multiple comparisons. We suggest that researchers adopt a combination of cluster-based permutation, hierarchical models, surrogate testing (*Cox and Fell, 2020*), and ROI to address the complexity of data structures across temporal, spectral, and spatial domains, and report both corrected and uncorrected results. Furthermore, we found that the majority of studies tend to selectively report results for significant electrodes, memory types, or coupling measures. These practices increase the likelihood of false positives and the overestimation of effect size (*Ioannidis et al., 2014*). We advocate for the standardization in reporting (1) all three coupling metrics, including coupling phase, strength, and prevalence, (2) their interactions with each other, and (3) their associations with memory performance. Each metric captures a distinct property of the coupling process and may interact with one another (*Weiner et al., 2024*), so it is necessary to provide a more comprehensive understanding of the coupling mechanism. We suggest that researchers should at least report exact values of effect sizes (e.g. correlation coefficients, standardized mean differences, or odds ratios), sample sizes, test statistics, p-values, and standard errors or confidence intervals for both significant and insignificant results to allow effective comparisons between studies. While only 2 out of the 23 studies included in our analysis disclosed all processed data and analysis code in their publication, we appreciate the responses of almost all authors who provided valid data or clarifications upon email requests that helped us mitigate heterogeneity and publication bias in subsequent analyses.

## Limitations and future research directions

The between-study discrepancy in measuring cross-frequency coupling presents a potential challenge to the comprehensive inclusion of studies. Due to methodological disparities and the limited prevalence of PETH studies, our current analysis only included studies using the preferred phase to measure the precision of coupling. Additionally, due to the expectation of high heterogeneity, we excluded gray literature without peer review and memory measures with low comparability to other studies, such as the targeted memory reactivation (*Hu et al., 2020*). While the inclusion of a restricted number of published studies may introduce a potential publication bias, we effectively addressed this concern through data requests, focal and sensitivity analysis, and meta-regression, finally constraining the overall risk of bias of most studies to a small to moderate level.

One of the advantages of our study is attributable to a substantial number of effect sizes and sample sizes. However, only a few studies reached a sample size of 30–50. In addition, our analysis

focused on studies with healthy populations. Future research should consider adopting more homogeneous analysis methods and openly sharing processed sleep and memory data, thereby aggregating sufficiently large and representative sample sizes in larger-scale studies. Moreover, our findings of no differences in the coupling-memory association between naps and overnight sleep suggest that nap paradigms may be efficient and thereby allow for a larger number of subjects.

Besides our suggestion for the classification of memory types and SP frequency, our results offer insights into studying causal sequences through more precise detection and stimulation of neural oscillations and offer insights for enhancing consolidation under aging or pathological conditions. We believe tracing dynamic distributions of SO-SP coupling and considering memory type and age-related regional changes would be intriguing and might be crucial for the understanding of the selective mediation of the spike-timing-dependent synaptic plasticity. High-density PSG has been instrumental in enhancing spatial sampling density for localizing SOs, subcortical areas, and even analyzing intra- and inter-regional PAC (*Massimini et al., 2004*; *Krishnaswamy et al., 2017*; *Gong et al., 2021*; *Riedner et al., 2007*; *Roehri et al., 2022*; *Routier et al., 2017*; *Seeber et al., 2019*; *Siebenhühner et al., 2016*). Moreover, integrating techniques with higher spatial resolutions, such as MRI and iEEG, will enhance the interpretative capacity of PSG results.

Finally, efforts must be made in future sleep research towards open science. To the best of our knowledge, as the first meta-analysis conducted on cross-frequency couplings, we have made our analysis code and data publicly available. Our posterior parameters predicted for each coupling measure can serve as priors in future meta-analyses, allowing for the integration of new data to update the model. We recommend that future sleep research should complete pre-registration on platforms such as OSF and clearly delineate pre-registered analyses from exploratory analyses in their reports to enhance methodological consistency and reproducibility. We also encourage original studies to share individual-level data and code. These efforts will contribute to enhancing reproducibility and comprehensive inclusion of results in future meta-analyses.

## Conclusion

This meta-analysis revealed the crucial role of the precise and strong SO-fast SP cross-frequency coupling in promoting memory consolidation with data demonstrating sufficient statistical power for the first time. By aggregating effect sizes of associations between four commonly used coupling measures and memory retention performance, we observed very strong evidence that the precision and strength of coupling can both predict enhanced memory retention scores across almost all conditions as a general physiological mechanism, while fast SP amplitudes detected in the frontal lobe also showed associations with memory consolidation. Although the effect sizes of the main models are relatively small, our moderator analyses demonstrated the dynamic nature of coupling-memory relationships. Stronger associations were observed in subgroups including young adults, frontal regions, fast spindles, and declarative memory, all of which have historically demonstrated relationships with better memory performance. Therefore, our results provide important information regarding complex memory consolidation mechanisms.

This evidence provides insights for future research on how the contribution of coupling to memory consolidation is modulated by behavioral and physiological factors, as well as meaningful information for the simultaneous implementation of MRI and electrical stimulation in the SO-SP coupling analysis. We believe that SO-SP coupling can offer more precise predictions for memory consolidation within the context of spatiotemporal specificity. This meta-analysis emphasizes the need for standardization and replication studies and points to those measures to focus on for future research in the field.

## Methods

### Literature search

The retrieval and screening of relevant studies were conducted in accordance with the 2020 PRISMA statement (*Page et al., 2021*). The PRISMA checklist is reported in *Supplementary file 1*. A comprehensive literature search was performed in three databases based on the retrieval qualities evaluation (*Gusenbauer and Haddaway, 2020*; *Martín-Martín et al., 2021*): PubMed, Web of Science, and PsycINFO, covering the period up to July 1, 2023. Only studies that have undergone peer review and have been published were included in the meta-analysis. Boolean operators were utilized to combine

the following search terms, (*sleep OR nap*) AND (*slow oscillat\* OR slow wave OR sleep oscillat\* OR slow-wave OR SO*) AND (*spindle OR sigma OR SP*) AND (*coupl\* OR pair\* OR lock\* OR coordinat\* OR interact\* OR synchro\**) AND (*motor learning OR memory OR cogniti\**). After filtering the papers based on inclusion criteria, a citation search was conducted and all papers identified from article citations underwent manual screening.

## Inclusion criteria

We applied the following criteria (**Meline, 2006**) to identify articles eligible for the meta-analysis as of July 1, 2023. Studies were included if the study (1) had memory encoding task before and recall measures after a sleep interval; (2) was published in English; (3) measured at least one of the standard SO-SP coupling measures (coupling phase, strength, percentage, and or SP amplitude) by phase-amplitude coupling (PAC) or comparable methods; (4) assessed the correlation between memory retention and SO-SP coupling during N2 and or SWS stage(s); (5) was published as an original research paper in a peer-reviewed journal. Studies were excluded if they (1) only measured SO-SP coupling influenced by the impact of medication or interventions (e.g. brain stimulation method) with no control condition; (2) only included non-human subjects, patient groups, or clinical groups (no control healthy group); (3) were a group under the wake condition. If a study included both groups that met

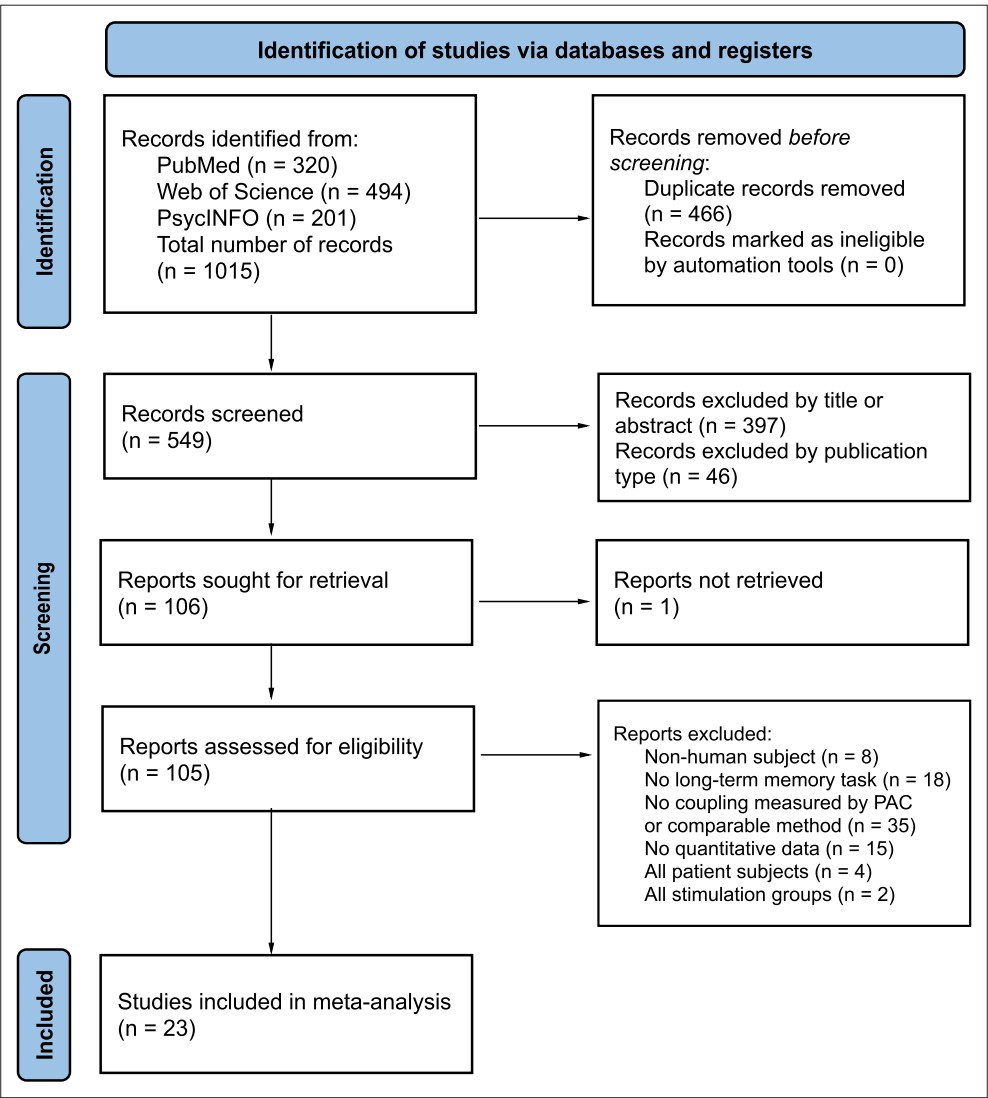

**Figure 9.** PRISMA flow diagram of literature search, screening, and inclusion for systematic review and meta-analysis.

the inclusion and exclusion criteria and those that did not, only data from the group(s) that met the criteria (e.g. control condition, control group) were included in the following analysis.

## Study selection

For the study selection, data retrieved from the three databases listed above were imported into EndNote 21 for automatic and manual duplication removal. The screening process is reported in *Figure 9*. After preliminary screening for titles, abstracts, and article types, 105 original research papers were screened by independent full-text review. Reviewers discussed all papers with discrepancies regarding their inclusion and ultimately achieved consistency. Finally, we included 23 eligible studies in the meta-analysis, comprising a total of 297 effect sizes from four types of coupling measurements with 730 samples.

## Data extraction

In the studies included in the meta-analysis, relevant effect sizes and specified study characteristics were extracted by two reviewers independently in accordance with PRISMA guidelines (*Page et al., 2021*) to minimize the bias introduced by subjective judgments. The consistency of data extraction between reviewers is 99.6%, and all discrepancies were resolved through discussion. Four specific measures of SO-SP coupling (*Figure 1*) were examined for their association with memory retention performance: (1) preferred coupling phase; (2) SP peak-to-trough amplitude; (3) coupling strength; and (4) coupling percentage. After extracting effect sizes, phase-radian alignment was corrected (*Figure 1C*). Extracted study characteristics included: (1) sample size, age, gender distribution; (2) pre-specified moderators (see Methodological characteristics and moderators for more details); (3) publication details. A summary of the main information for each study is outlined in *Appendix 2— table 2*. The detailed data for each effect size can be found in *Source data 1* (study-level), source data of *Figures 4, 6–8* (effect size-level), and the publicly available repository, https://osf.io/9mh5d/. The overall quality of studies is assessed in accordance with the Robins-I (*Jüni, 2016*; *Sterne et al., 2016*) and NIH criteria (*National Heart, Lung, and Blood Institute, 2019*), with adjustments (detailed in *Appendix 1—table 1*) made to accommodate the specific attributes of the study.

Due to the relatively large heterogeneity and discrepancies in methodology and results reported between studies, as well as the non-parametric nature of the direction in circular-linear correlations, we improved comparability and credibility across studies by requesting both unreported effect sizes and processed individual-level memory and physiological data to reduce the publication bias. If a study measured relevant sleep and memory features but either (1) did not report any extractable or convertible effect sizes; (2) reported imprecise p-values due to insignificance; (3) reported only one effect size for cross-group data; (4) reported effect sizes that were not comparable to other studies, coupled with the absence of data in supplementary material, we requested missing data or processed individual data from corresponding authors via email. 21 out of 23 authors responded positively and provided the requested data. Studies without a response after two months or declined our requests were excluded or partially excluded from the meta-analysis for the part where insufficient information was provided for calculating effect sizes. For studies reporting correlations only through scatterplots, linear correlations were estimated using software from the ShinyDigitise package (*Pick et al., 2019*; *Ivimey-Cook et al., 2023*) in R, while circular-linear correlations were first assessed by extracting estimated individual data from the plots by the online software WebPlotDigitizer (*Rohatgi, 2014*), followed by the circular-linear correlation analysis in R. Additionally, key missing study characteristic data was requested via email at the same time to broaden the scope of moderator analysis.

## Effect size calculations

To standardize effect sizes for comparability across studies, we chose to standardize the bounded circular-linear correlation coefficient (*Mardia, 1976*; *Fisher, 1995*) to unbounded $r_z$ to examine the association between coupling phase (in radians) and memory consolidation. We used the Pearson product-moment correlation coefficient *r* (*Lipsey and Wilson, 2001*) (hereinafter referred to as Pearson's *r*) and transformed Fisher's *z* to report the linear relationships between SP amplitude (in μV), coupling strength (mean vector length or modulation index), coupling percentage (%), with memory consolidation separately. All transformable effect size measures including *t*-statistic, *p*-value, $\beta$ statistic, and $\eta^2$, were extracted and converted to Pearson's *r*. In cases where effect measures were reported as

non-parametric correlation including Spearman's rho ($\rho$, i.e. $r_s$) and no author response could reanalyze for parametric correlation, we used it as an imperfect estimation of Pearson's $r$ (**Myers and Sirois, 2014**) and excluded from sensitivity analyses.

## Pearson product-moment correlation coefficient

The constrained range of values between −1 and 1 measured by Pearson's $r$ restricts the selection of an ideal unbounded prior distribution for Bayesian hierarchical models (**Thompson and Semma, 2020**; discussed in Statistical analysis). The deviation of its sampling distribution from the assumptions of normal, especially when the effect size is large, might provide an inaccurate estimation for the sampling variance (**Fisher, 1921**). To address this limitation, we used the metafor package (**Viechtbauer, 2010**) in R to transform Pearson's $r$ into normalized and unbounded Fisher's $z$ by formula (**Fisher, 1921**):

$$z = \frac{1}{2} \ln\left(\frac{1+r}{1-r}\right) = \tanh^{-1} r$$

When reporting and interpreting the results, we reversed the transformation from Fisher's $z$ back to Pearson's $r$ to present each estimated effect size and credible interval (CrI) by formula:

$$r = \left(\frac{e^{2z}-1}{e^{2z}+1}\right) = \tanh z$$

## Standardized circular-linear correlation coefficient

Meanwhile, the circular-linear correlation coefficient $r$ (hereinafter referred to as *circlin r*) is a type of Pearson correlation coefficient (PCC) distributed from 0 to 1 without direction, achieved by transforming the phase into $\sin\theta$ and $\cos\theta$ to create linear parameters (**Mardia, 1976**; **Fisher, 1995**), to assess the relationship between a random unit vector and another linear random variable:

$$circlin\ r = \sqrt{\frac{r_{12}^2 + r_{13}^2 - 2 \cdot r_{12} \cdot r_{13} \cdot r_{23}}{1 - r_{23}^2}}$$

$$r_{12} = \text{corr}(x, \cos\theta), \ r_{13} = \text{corr}(x, \sin\theta), \ r_{23} = \text{corr}(\cos\theta, \sin\theta)$$

Since there is no existing method to standardize the *circlin r*, we developed an approximation approach to transform the bounded and non-linear distribution to an unbounded standardized normal distribution $N(0,1)$. The verification and performance can be found in Appendix 5.

The population distribution of circular-linear correlation $\rho$ with $n$ samples can be approximated by a chi-square distribution with 2 degrees of freedom (**Fisher et al., 1993**), $\chi_2^2$. We derived that the circular-linear correlation coefficient has the following mean and variance in an approximate form and verified in R,

$$\mathbb{E}(\rho) = \sqrt{\frac{\pi}{2n}}, \ \ \text{Var}(\rho) = \frac{4-\pi}{2n}, \ \ 0 \leq r \leq 1, \ n \geq 2$$

We can observe that different from Pearson's $r$, which always has a population mean $H_0 : \rho = 0$ under the null hypothesis, the *circlin r* has a population mean between 0 and 1, which tends to approach 1 and display left-skewness as the sample size $n$ decreases, while approach 0 and exhibiting right-skewness as the $n$ increases (also see **Appendix 5—figure 1**). The sampling distribution of correlated *circlin r* also shows the same property of skewness when approaching lower and upper bounds (**Appendix 5—figure 2**). Therefore, it is clearly not appropriate to use raw coefficients in the meta-analysis.

To transform its sampling distribution to be unbounded, normally distributed, and centered at 0 for our meta-analysis, we first scaled the *circlin r* as a form of $nr^2$ to approximate the $\chi_2^2$ distribution (**Fisher et al., 1993**). Here, the sample size has been weighted to eliminate the bias introduced by large $r$ under small sample sizes. We can find the quantile of *circlin r* in the upper tail,

$$\Pr\left(nr^2 \leq n\rho^2\right) = 1 - e^{-nr^2/2}, \ \ 0 \leq r < 1$$

**Table 6.** Interpretation of the standardized circular-linear correlation coefficient.

| $r_z$ | Strength |
|---|---|
| <0 | No Effect |
| 0 | Null |
| 0.1 | Small |
| 0.3 | Moderate |
| 0.5 | Strong |

Finally, by transforming non-normal chi-square deviates to standardized normal distribution and scaling the distribution by the sample size, *circlin r* can be transformed into a standardized normal scale $r_z$,

$$\Phi\left(r_z\sqrt{n}\right) = \mathrm{Pr}\left(nr^2 \leq n\rho^2\right) = \frac{1}{\sqrt{2\pi}} \int_{-\infty}^{r_z\sqrt{n}} e^{-y^2/2}\, dy$$

The sampling distribution of the transformed *circlin* $r_z$ approximates a normal distribution when $n > 15$. For $n < 15$, the distribution is slightly right-skewed. Its population distribution can be approximated by Pearson's *r*, and it is bounded between −1 (null) and 1 (alternative). The interpretation of the strength of $r_z$ is comparable with Pearson's *r* (see *Table 6*), and now $r_z$ can also be transformed to Fisher's *z* and included in the meta-analysis. It is worth noting that small effect sizes are common in neuroscience and meta-analyses due to the complexity of underlying mechanisms and the presence of numerous confounding variables and hierarchical structures, so small correlations may carry substantial and meaningful information to interpret. Monte Carlo simulation results and the code used for transformations were also reported in Appendix 5.

## Methodological characteristics and moderators

In previous studies, researchers have applied different experimental designs to investigate coupling and summarized divergent conclusions regarding each coupling parameter. To develop a more systematic understanding of the heterogeneity of approaches, six moderators were selected in the moderator analysis. *Appendix 2—table 2* provides a complete list of moderators in each study, and categories that define moderators are listed below, the range represents the number of effect sizes included in the analysis across four types of coupling measures:

**Table 7.** Summary of memory tasks.

| Memory Task Domain | Memory Task Modality |
|---|---|
| | Word-Pair (WP) |
| | Word List |
| | Novel Metaphor |
| Verbal Tasks (*n*=12) | Word-Image Pair (WIP) |
| Emotional Task (*n*=3) | Picture-recognition (IMG) |
| | Spatial Memory |
| | 2D Object Location (2DL) |
| Spatial Tasks (*n*=4) | Visuo-spatial (VS) |
| | Motor Sequence (MST) |
| | Gross-motor Juggle |
| | Mirror-tracing Task (MTT) |
| Motor Tasks (*n*=6) | Visuomotor Adaptation (VMA) |

1. **Sleep Stage:** Sleep stages were grouped by N2 (nREM2; $k = 11 - 18$) and SWS (slow-wave sleep; $k = 9 - 30$), while uncategorized sleep stages were encoded as 'mixed' ($k = 23 - 49$).
2. **Sleep Bout:** Nighttime sleep lasting typically more than 6 hours was encoded as 'overnight' ($k = 32 - 73$), while short naps lasting around 1–2 hr during the early morning or afternoon were classified in the 'nap' group ($k = 11 - 27$).
3. **Age:** Due to the limited number of studies on children, adolescents, and older adults, age was encoded as a continuous variable based on the mean age of each study for the meta-regression.
4. **PSG Channel:** Effect size has been categorized into three clusters based on the cortical area under PSG electrodes, including frontal (F3, Fz, and F4; $k = 15 - 36$), central (C3, Cz, and C4; $k = 13 - 29$), and posterior (P3, Pz, P4, O1, and O2; $k = 8 - 22$), and mixed ($k = 3 - 7$). Other electrodes within the same area but exhibiting excessive deviation from the midline or containing insufficient information have been excluded (e.g. C1, PO8). Sleep parameters were computed by averaging SP amplitude, coupling phase, strength, and percentage across electrodes within each cluster.
5. **Spindle Type:** To include a wider range of comparable studies, we specified frequency boundaries for fast SPs (12–16 Hz; $k = 22 - 43$) and slow SPs (8–12 Hz; $k = 16 - 28$). Effect sizes that transversed frequency boundaries or reported together are classified as 'mixed' ($k = 16 - 28$).
6. **Memory Type:** We classified comparable memory tasks into four domains presented in *Table 7*. Verbal memory can also be interpreted as neutral non-spatial memory, in contrast to the other two types of declarative memory.

## Statistical analysis

We chose to fit overall and subgroup (moderator) models using Bayesian hierarchical random-effects and mixed-effects models, respectively, due to the common occurrence of multiple effect sizes for the same set of participants reported within the same study, violating the assumption of independence of effect sizes, as well as the limited and unequal group size and potential high heterogeneity. In comparison to frequentist models, Bayesian models incorporate prior probabilities for each parameter by considering likelihood information (i.e. the effect sizes we extracted) to establish a model for predicting the posterior probability distribution of parameters, which provides transparent inferences with lower risks of false positives. In the posterior distribution, confidence intervals (CIs) reported by frequentist methods are replaced by credible intervals (CrI), which can be interpreted as 'there is a 95% probability that the parameter lies within the interval', thereby providing a more precise prediction of true probabilities. Additionally, Bayesian methods could more effectively build models that account for multiple sources of heterogeneity and allow the analysis of the impact of moderator variables representing different measures and participant groups on the pooled effect size (*Sutton and Abrams, 2001*). As *Kruschke and Liddell, 2018* described, the shrinkage property of Bayesian models helps prevent false alarms by pulling extreme values toward the group mean, which is especially valuable when accounting for potentially outlying estimates. This explains the observed differences between model distributions and the raw effect sizes in our forest plots. All statistical analyses for Bayesian models were conducted in R using the brms package (*Bürkner, 2017*) along with supporting packages bayesplot (*Gabry et al., 2019*), metaviz (*Wagner et al., 2018*), and customized codes for visualization. Contrary to the misconception that Bayesian models are overly complex or opaque, they are increasingly valued for their accuracy and transparent inferences (*Kruschke and Liddell, 2018*). We also recognize that some researchers may prefer frequentist approaches. To support transparency and comparability, we also provided the traditional meta-analytic results in *Supplementary file 5*, which demonstrate consistency with our Bayesian findings.

For each overall and subgroup model (*Appendix 2—table 1*), we applied the Markov chain Monte Carlo (MCMC) method, a general family of algorithms used to approximate complex probability distributions, with the no-U-turn Hamiltonian Monte Carlo (HMC) sampler, the default algorithm in Stan, to set up four chains, with each chain undergoing 12,000 iterations (including 2000 warm-ups), the minimum requirement for calculating accurate Bayes Factors (BF; *Kass and Raftery, 1995*), which reflects how much more likely the data are under one hypothesis. The target average acceptance probability was set to 0.99 with the maximum tree depth to 15 to ensure robustness in the posterior distribution.

Convergence of the MCMC was checked following the suggestion of WAMBS-Checklist (*Depaoli and van de Schoot, 2017*) through 1. graphical posterior predictive checks to evaluate how well

the model-predicted data replicated the observed data; 2. trace plots; and 3. the Gelman-Rubin diagnostic to assess convergence of the Markov chains. Ideally, the posterior distribution should overlap with the distribution of the test data generated, the trace plot distribution should resemble a uniformly undulating wave with high overlap between chains, and the Potential Scale Reduction Factor (R̂) should be less than 1.1. Autocorrelation plots were used to ensure low temporal dependency between successive samples. Examples of diagnostic plots were reported in **Supplementary file 2**. Any non-convergence at the aforementioned stages led to reconfiguration of chains and iterations for analysis. Estimation of each intercept, moderator, and heterogeneity in the posterior distribution was extracted, and the estimation of mean, distribution, and 95% credible interval of each effect size and the pooled effect size were reported in forest plots.

## Overall model

We fitted random-effects models that predict the relationships between coupling phase, SP amplitude, coupling strength, and coupling percentage with memory consolidation, considering only heterogeneity and sampling errors. The three-level Bayesian model superimposed random effects for sampling error (first level), between-study heterogeneity (second level), and within-study heterogeneity (third level):

$$z \mid \sigma_z \sim 1 + (1 \mid \text{Effect}) + (1 \mid \text{Study})$$

By introducing priors and likelihood information to model intercept and heterogeneity parameters, our random-effects model can be represented using the following formula:

$$\hat{\theta} \sim N(\theta_{ij}, \sigma_{ij}^2)$$

$$\theta_{ij} \sim N(\theta_i, \sigma_i^2)$$

$$\theta_i \sim N(\mu_\theta, \tau^2)$$

$$\hat{\theta} \sim N(\mu_\theta,\ \sigma_{ij}^2 + \sigma_i^2 + \tau^2)$$

$$(\mu_\theta, \tau^2, \sigma_i^2) \sim p(\cdot), \quad \tau^2, \sigma_i^2 > 0$$

$$\mu_\theta \sim N(z_{\text{mean}}, z_{\text{sd}})$$

$$\tau, \sigma_i \sim HC(z_{lp}, z_{sp})$$

where z represents the fisher's *z*-transformed correlation coefficient, $\tau^2$ represents between-study heterogeneity, $\sigma_i^2$ represents within-study heterogeneity, and $\sigma_{ij}^2$ represents sampling error. Since the distribution of Fisher's *z* and heterogeneity (**Röver et al., 2021**) all follow certain probability functions, and non-informative priors could not provide reasonable estimates, we selected weak informative priors in the model to trade off between allowing collected real data to influence posterior distributions and excluding extreme outliers. For the intercept of all models, we chose a standardized normal distribution *N*(0,1) as the prior distribution, assuming no effect to control false positives.

The prior for heterogeneity used the Half-Cauchy distribution, known for its heavy-tailed property, improving it to be more tolerant of high heterogeneity (**Williams et al., 2018**). Also, half-Cauchy truncated at 0, which is consistent with the fact that heterogeneity cannot be less than 0. For between-study heterogeneity $\tau^2$, we obtained a mean between-study heterogeneity of $\mu = 0.13$ from a dataset of heterogeneity reported in 498 correlational meta-analyses (**Van Erp et al., 2017**) using Fisher's *z* or Pearson's *r* as the measure of effect size, which represents a replication of the methodology by **McKinney et al., 2021** regarding prior selection. Additionally, due to differences in measurement methods and memory tasks across studies, we held a prior belief in some degree of between-study heterogeneity. For reasons discussed above, we set the prior using half-Cauchy distribution $HC(0, 0.5)$ by extending the scale parameter from 0.13 to 0.5 to account for more extreme heterogeneity. Lastly, the prior for within-study heterogeneity is also set as to be $HC(0, 0.5)$ to balance the uncertainty arising from different measurement approaches or sample groups. This combination of priors reduced the risk of overestimation, accounted for substantial uncertainty, and increased transparency by explicitly encoding all assumptions.

**Table 8.** Interpretation of Bayes Factor ($BF_{10}$) for the strength of evidence.

| $BF_{10}$ | Direction |
| --- | --- |
| <0.1 | Strong, Favor alternative |
| 0.1–0.33 | Moderate, Favor alternative |
| 0.33–1 | Weak, Favor alternative |
| 1–3 | Weak, Favor hypothesis |
| 3–10 | Moderate, Favor hypothesis |
| >10 | Strong, Favor hypothesis |

## Moderator models

For moderator analysis, we introduced continuous and categorical variables into the overall model as fixed effects, forming a following mixed-effects model:

$$z \mid \sigma_z \sim 1 + \text{Moderator} + (1 \mid \text{Effect}) + (1 \mid \text{Study})$$

$$\hat{\theta} \sim N\left(\mu_\theta + \sum_{k=1}^{n} \beta_k \cdot X_{ijk}, \ \sigma_{ij}^2 + \sigma_i^2 + \tau^2\right)$$

in which $X$ denotes random variables for moderators, $k$ is the random variable of moderators, and $n$ represents the number of moderators specified in the model. In the subgroup model, we chose to use the weak informative prior $N(0, 1)$ for each moderator as fixed effects to account for the uncertainty in the magnitude of the aggregate effects caused by moderators. In the meta-regression model accounting for the effect of age, we chose $N(0, 0.1)$ to model the change over years. Except for the exploratory analysis, all statistical models used in data analysis are summarized in *Appendix 2—table 1*. All posterior distributions of the moderators are reported in *Supplementary file 6*.

To assess directional hypotheses proposed for each moderator model, we conducted non-linear hypothesis testing within the Bayesian framework, obtaining Bayesian Factors ($BF_{10}$, defined as Evidence Ratio, ER, in the package brms) for different levels within each model, which performed tests of evidence for the ratio of marginal likelihoods between two levels. In addition, we conducted hypothesis testing on the overall model compared to the null. *Table 8* presents the strength of evidence represented by different Bayesian Factors, which is analogous to the one-tailed t-test in the interpretation. In directional hypotheses, $BF_{10} = 3$ means the hypothesis is three times more likely than the alternative. Additionally, we used the posterior probability, also defined as credibility score (*Cox et al., 2023*), to report the percentage of posterior samples consistent with the direction of the hypothesis. A posterior probability of 1 indicates that all posterior sample draws align with the hypothesis.

The predictive power of models was compared in pairs by computing model weights via leave-one-out (LOO) stacking of posterior predictive distributions (*Vehtari et al., 2017*), including the comparison between the overall model and a single moderator model, as well as a single moderator model and a moderator model with an additional moderator and interaction term. Models with additional predictor(s) that had higher stacked weights indicate that it enhanced the overall predictive power of the model. This approach could help identify moderators that influence the correlation between SO-SP coupling and memory retention. Model weights have also been computed via the Pseudo-BMA method as the controlled analysis.

## Sensitivity analysis

Finally, in sensitivity testing, we conducted separate assessments for (1) publication bias; (2) focal models excluding studies that introduced significant heterogeneity; and (3) the impact of priors. Regarding publication bias, although efforts were made to minimize potential controllable biases through the data request (especially for effect size computed but not reported by studies), effect size transformation, and multi-level models, differences in data analysis approaches and experimental conditions could not be fully addressed at the meta-analysis level. Therefore, we additionally used a frequentist approach by using Restricted Maximum Likelihood (REML) with the metafor package

(*Viechtbauer, 2010*) to fit three-level models to extract data for generating funnel plots and quantifying publication bias. Frequentist results of each model were also reported in *Supplementary file 5* for readers unfamiliar with Bayesian statistics. Besides the Egger's regression test and rank correlation test, we chose counter-enhanced funnel plot (*Kossmeier et al., 2019*) and superimposed the Egger's regression line. We also performed the time lag bias test to quantify the impact of heterogeneity and effects of the year of publication.

In the focal analysis, we excluded studies focused only on non-declarative memory tasks, those with a high risk of bias (see Appendix 1) or significant methodological differences, those only tested post-sleep memory retention, or with a Pareto $k$ diagnostic value larger than 0.7. Pareto $k$ for each effect size is reported in *Supplementary file 4*. We also removed studies that adopted modulation index as their measures in the focal model of coupling strength. In addition, the method of prior sensitivity test, as well as the posterior predictive check, was reported in *Supplementary file 2*.

## Phase spatiotemporal analysis

The non-negative and nonlinear distribution of circular-linear correlation makes it impossible to determine the direction of correlation. This is one of the reasons why we have requested individual-level processed data from most of the authors of included studies, in addition to requesting the effect size. By standardizing memory scores of each individual in each study using $z$-scores, we overlaid the data from comparable studies to fit and visualize the nonlinear relationship between the coupling phase and memory consolidation using the best fit second degree quadratic line for the combination of each SP type and channel location. It could effectively improve the interpretation of nonlinear relationships and circular direction of the SO-SP coupling phase across studies.

For the comparison of coupling phase across PSG channel clusters, we used the Bayesian circular mixed-effects model (*Cremers and Klugkist, 2018*) with 12,000 iterations to account for repeated measurement and differences of sample sizes between channels. PSG channel location has been set as a fixed effect, while subject has been taken into account as a random effect. A 95% credible interval and the posterior distribution of each circular direction was reported in the circular plot to compare the timing of occurrence of SO-SP coupling across different cortical areas. Similar to the main analysis, the Bayes factor and posterior probability have been used to evaluate evidence for the circular mean difference. In addition, the inconsistency of the coupling phase has been assessed by the Rayleigh test.

# Acknowledgements

We would like to express our sincere gratitude to all authors of included studies who have generously shared demographic, physiological, behavioral data, or effect sizes, or clarified experimental methods with us through email communication. Work of RMCS was funded in part by NIH R01 AG040133.

# Additional information

### Funding

| Funder | Grant reference number | Author |
| --- | --- | --- |
| National Institutes of Health | R01 AG040133 | Rebecca MC Spencer |

The funders had no role in study design, data collection and interpretation, or the decision to submit the work for publication.

### Author contributions

Thea Ng, Conceptualization, Data curation, Software, Formal analysis, Validation, Investigation, Visualization, Methodology, Writing – original draft, Writing – review and editing; Eunsol Noh, Conceptualization, Methodology; Rebecca MC Spencer, Conceptualization, Resources, Funding acquisition, Writing – original draft, Writing – review and editing

## Author ORCIDs

Thea Ng 🔗 https://orcid.org/0009-0008-9521-9254
Rebecca MC Spencer 🔗 https://orcid.org/0000-0002-8674-2384

Reviewer #1 (Public review): https://doi.org/10.7554/eLife.101992.3.sa1
Reviewer #2 (Public review): https://doi.org/10.7554/eLife.101992.3.sa2
Reviewer #3 (Public review): https://doi.org/10.7554/eLife.101992.3.sa3
Author response https://doi.org/10.7554/eLife.101992.3.sa4

## Additional files

### Supplementary files

MDAR checklist

Supplementary file 1. PRISMA statements checklist.

Supplementary file 2. Model diagnostics. Example plots demonstrating the diagnostic process for each fitted Bayesian model, including posterior predictive checks, trace plots, and autocorrelation plots.

Supplementary file 3. Effect size-level forest plot.

Supplementary file 4. Pareto $k$ diagnostic statistics.

Supplementary file 5. Frequentist analysis. Forest plots generated using frequentist analysis for each fitted overall Bayesian model.

Supplementary file 6. Posterior distributions of moderators.

Source data 1. Main table of study characteristics and participant demographics.

### Data availability

All effect size-level and study-level data used in this meta-analysis are available in the Open Science Framework repository at https://osf.io/9mh5d/, and shared in the source data.

The following dataset was generated:

| Author(s) | Year | Dataset title | Dataset URL | Database and Identifier |
|---|---|---|---|---|
| Ng T, Noh E, Spencer R | 2024 | Does slow oscillation-spindle coupling contribute to sleep-dependent memory consolidation? A Bayesian meta-analysis | https://doi.org/10.17605/OSF.IO/9MH5D | Open Science Framework, 10.17605/OSF.IO/9MH5D |

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

## Appendix 1

### Risk of bias assessment

Due to the absence of a dedicated risk of bias assessment tool or guideline specifically designed for meta-analysis of correlation coefficients, we made minor adaptations based on established criteria and signaling questions from the Risk Of Bias In Non-randomized Studies of Interventions (Robins-I) framework (*Jüni, 2016*; *Sterne et al., 2016*). These adaptations entailed substituting descriptions related to interventions and control experiments with references to sleep and single-group experiments, omitting criteria that were not applicable to the specific objectives of our study, adjusting domain classifications, and including standards from the NIH Study Quality Assessment Tools for Before-After (Pre-Post) Studies With No Control Group (*National Heart, Lung, and Blood Institute, 2019*) in the evaluation of certain studies. In addition, we have augmented our assessment comprehensively by introducing supplemental signaling questions, as outlined in *Appendix 1—table 1*, in accordance with the research methods and data extraction procedures detailed in the meta-analysis. For the dataset we requested from the author, it would still be marked as low risk when the above conditions were met. The full version of signaling questions can be found in Robins-I. The outcomes of the risk of bias assessment for each research paper included in the meta-analysis can be found in *Appendix 1—figure 1*, generated by the robvis package (*McGuinness and Higgins, 2021*) in R.

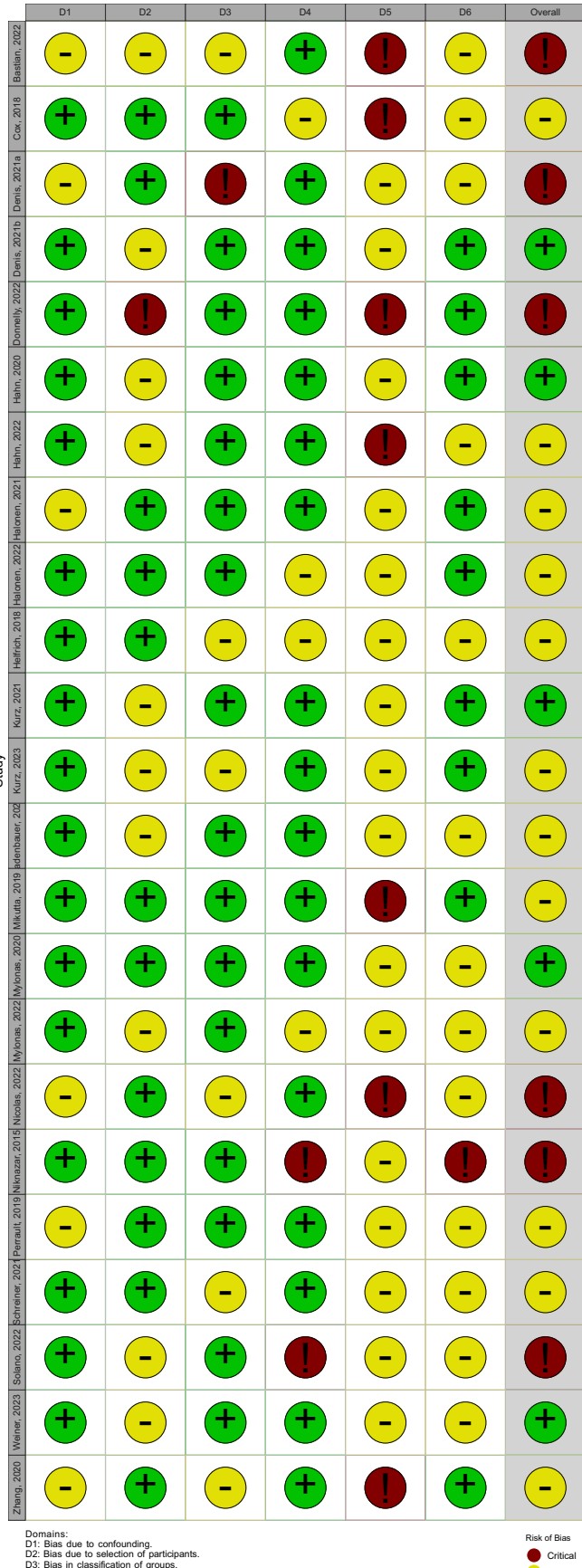

Domains:
D1: Bias due to confounding.
D2: Bias due to selection of participants.
D3: Bias in classification of groups.
D4: Bias due to missing data.
D5: Bias in measurement of outcomes.
D6: Bias in selection of the reported result.

Risk of Bias
● Critical
● Moderate
● Low

**Appendix 1—figure 1.** Risk of bias (ROB) assessment for individual studies.

**Appendix 1—table 1.** Risk of bias assessment supplemental criteria.

| Domain | Supplemental Signaling Questions |
| --- | --- |
| Bias due to confounding variables | 1.1 Was sufficient information provided to assess the presence of major potential confounding variables? |
| | 1.2 Were major potential confounding variables not relevant to study controlled during the data collection and analysis? Were influences from other experimental tasks or stimuli existing? |
| | 1.3 Were confounding factors (such as gender, age, etc.) added to models to calculate and interpret as the "corrected" effect size? |
| Bias due to subject selection | 2.1 Was subject selection representative? Can subjects represent the population or community targeted by the experiment? |
| | 2.2 Was a random sampling method applied during data collection? Are subjects recruited mostly from a single source (e.g., university) or at different times and caused biases? |
| | 2.3 Are the subjects independent from each other? Are the subjects socially connected (e.g., patients and their relatives, between groups if there are multiple groups in the original study)? Have the same subjects been measured repeatedly in pretest-posttest designs? |
| Bias due to classification of groups | 3.1 For studies with multiple groups, can group(s) containing only healthy human subjects without intervention be clearly classified? |
| | 3.2 When reporting the effect size, did the authors report the effect size separately for different groups? |
| Bias due to missing outcome data | 4.1 Were only scatterplots reported in the article, necessitating the use of graph tools to estimate the effect size? |
| | 4.2 Was data including $t$ statistics, $p$-values, $\beta$ statistics, and $\eta^2$ the only data provided that could be used to estimate the correlation, or only imprecise data provided for non-significant correlation? |
| | 4.3 Have pre-sleep/sleep/post-sleep memories and/or sleep data for individual participants been lost? If true, were missing outcome data interpolated, averaged, simulated, or deleted? |
| Bias in measurement of the outcome | 5.1 Were measurements, units, and signal processing approaches (including slow oscillillation, spindle, and coupling detection) used by the paper reliable and consistent with others? |
| | 5.2 Were only nonparametric effect sizes including Spearman's rho ($\rho$), Kendall's tau ($\tau$), or subjective estimation reported instead of Pearson's $r$, Fisher's $z$, or circular linear $r$? |
| | 5.3 Did the authors analyze, transform, or clarify non-normal data, introduce resampling techniques and/or exclude outliers? |
| Bias in selection of the reported result | 6.1 For multiple groups measured, if only group(s) with the largest effect size, or supported their hypotheses were reported, while non-significant or contradictory results were omitted? |

*Appendix 1—table 1 Continued on next page*

*Appendix 1—table 1 Continued*

| Domain | Supplemental Signaling Questions |
|---|---|
| | 6.2 Have the authors declared their research proposal and expected outcomes within the framework of pre-registration, or declared any conflict of interest in the paper? |
| | 7.1 Does the article not fully meet the applicable expectations of the Risk Of Bias In Non-randomized Studies of Interventions (Robins-I) and the above supplementary criteria in multiple domains listed? |
| Overall Bias | 7.2 Alternatively, does the article significantly conflict with the criteria in one of these domains? |

## Appendix 2

## Summary of models and included datasets

Appendix 2—table 1. Summary of models for SO–SP coupling–memory association measures.

| Model | Moderator | Model Purposes |
| --- | --- | --- |
| M | None | Study associations between SP amplitude, coupling phase, coupling strength, and coupling percentage, each in relation to memory consolidation. |
| M1 | Memory Task | Investigate potential distinctions in coupling and memory association mechanisms between declarative memory—including verbal, spatial, and emotional memory—as well as procedural memory. |
| M2 | Mean Age | Understand the potential impact of development and aging in coupling and memory associations. |
| M3 | Spindle Type | Explore how the coupling between SOs and fast or slow SPs predicts memory retention performance differently. |
| M4 | PSG Channel | Study the relation between sleep brain oscillations and memory in different cortical regions, as the frontal, central, and parietal areas were reported to be the most active area for SO–SP coupling but might play different roles. |
| M5 | Sleep Stage | Examine the impact of sleep stage on the relationship between SO–SP coupling and memory, considering that SPs are most active during N2 sleep, while SOs dominate cortical oscillation during SWS. |
| M6 | Sleep Bout | Investigate the potential impact of sleep timing and circadian rhythms on the relationship between SO–SP coupling and memory. |
| M7 | Age × Channel | Investigate interactions between age differences and PSG channels in the memory consolidation mechanism, considering the frontal lobe is the latest area of the brain to develop. |
| M8 | Age × Task | Study whether age increase implies a difference in predictive power of SO–SP coupling in the development of declarative and procedural memory consolidation. |
| M9 | Channel × Spindle | Examine interactions between SP types and cortical areas in models. |
| Mf | All predictors | Include all pre-specified moderators to the model as fixed effects to explore their explanatory power and potential collinearity. |
| Mc | None | Fitted controlled model for focal and sensitivity analysis. |

Appendix 2—table 2. Main characteristics for each study included in the meta-analysis.

| Author (Year) | N | $M_{Age}$ | Stage | Condition | PSG Channels | Spindles | Task | Memory Type | Measures Included |
| --- | --- | --- | --- | --- | --- | --- | --- | --- | --- |
| *Bastian et al., 2022* | 15 | 23.3 | N2, SWS | nap | frontal, central | fast, slow | Spatial | declarative | phase, amp, str, pct |

*Appendix 2—table 2 Continued on next page*

*Appendix 2—table 2 Continued*

| Author (Year) | N | $M_{Age}$ | Stage | Condition | PSG Channels | Spindles | Task | Memory Type | Measures Included |
|---|---|---|---|---|---|---|---|---|---|
| *Cox et al., 2018* | 24 | 30.2 | N2, SWS | overnight | frontal, central, posterior | fast, slow | MST | motor | phase, str |
| *Denis et al., 2022* | 31 | 22.3 | N2, SWS | overnight | frontal, central | fast | IMG | declarative | phase, str, pct |
| *Denis et al., 2021* | 34 | 22 | N2, SWS | nap | frontal, central, posterior | fast, slow | WP | declarative | phase, amp, str, pct |
| *Donnelly et al., 2022* | 16 | 14.1 | N2, SWS | overnight | frontal, central, posterior | fast, slow | 2 DL | declarative | phase, amp, str |
| *Hahn et al., 2020* | 33 | CH: 9.5, AD: 16 | SWS | overnight | frontal, central, posterior | fast, slow | WP | declarative | phase, amp, str, pct |
| *Hahn et al., 2022* | 42 | AD: 12.9, YA: 22.0 | SWS | overnight | frontal, central, posterior | fast, slow | Juggle | motor | phase, amp, str |
| *Halonen et al., 2021* | 27 | 22 | N2, SWS | overnight | frontal, central | fast | Metaphor | declarative | phase, str, pct |
| *Halonen et al., 2022* | 151 | 17 | N2, SWS | overnight | frontal, central | fast, slow | IMG | declarative | phase, amp, str, pct |
| *Helfrich et al., 2018* | 52 | YA: 20.4, OA: 73.8 | SWS | overnight | frontal | fast | WP | declarative | phase, str |
| *Kurz et al., 2021* | 19 | 11.24 | N2, SWS | overnight | frontal, central, posterior | fast, slow | IMG | declarative | phase, amp, str, pct |
| *Kurz et al., 2023* | 30 | 11.43 | N2, SWS | overnight | frontal, central, posterior | fast, slow | Word List | declarative | phase, amp, str |
| *Ladenbauer et al., 2021* | 43 | YA: 23, OA: 66 | N2, SWS | nap | frontal, central | fast | WP, VS | declarative | phase, amp, str |
| *Mikutta et al., 2019* | 20 | 27.1 | N2, SWS | overnight | central | fast, slow | Word List, MTT | declarative, motor | phase, amp, str |
| *Mylonas et al., 2020* | 28 | 30 | N2 | overnight | central | fast | MST | motor | phase, amp, str, pct |
| *Mylonas et al., 2022* | 14 | 13 | N2 | overnight | central | fast, slow | Spatial | declarative | phase, amp, str, pct |
| *Nicolas et al., 2022* | 24 | 21.9 | N2, SWS | nap | frontal, central, posterior | fast | MST | motor | phase, amp |
| *Niknazar et al., 2015* | 28 | 22 | N2 | nap | central | fast | WP | declarative | phase, amp, str |
| *Perrault et al., 2019* | 16 | 23.4 | N2, SWS | overnight | frontal, posterior | fast, slow | WP | declarative | phase, amp |
| *Schreiner et al., 2021* | 20 | 20.8 | N2, SWS | nap | central | fast | WIP | declarative | phase, amp, str, pct |

*Appendix 2—table 2 Continued on next page*

*Appendix 2—table 2 Continued*

| Author (Year) | N | $M_{Age}$ | Stage | Condition | PSG Channels | Spindles | Task | Memory Type | Measures Included |
|---|---|---|---|---|---|---|---|---|---|
| *Solano et al., 2022* | 10 | 24.3 | N2, SWS | overnight | frontal, central, posterior | fast, slow | VMA | motor | phase, amp, str, pct |
| *Weiner et al., 2024* | 25 | 69.1 | N2, SWS | overnight | frontal, central | fast, slow | WP | declarative | phase, amp, str |
| *Zhang et al., 2020* | 28 | 20.6 | N2, SWS | overnight | frontal, central, posterior | fast, slow | WP | declarative | phase, str |

Notes. *N* sample size, only includes groups in the meta-analysis; *N2* Stage 2 nREM sleep; *SWS* slow wave sleep; *amp* spindle amplitude; *str* coupling strength; *pct* oupling percentage; *CH* children; *AD* adolescents; *YA* young adults; *OA* older adults; PSG posterior channels include both parietal and occipital electrodes. Details about memory task types are listed in **Table 7**; additional detailed information and data are listed in **Source data 1** (study-level characteristics) and source data of **Figures 4 and 6–8** (effect size-level characteristics of each coupling metric).

# Appendix 3

## Interaction models and sensitivity analysis

**Appendix 3—table 1.** Summary of interaction and sensitivity models for the coupling phase-memory association.

| Models | Weight | Factors | Estimate (95% CrI) | Age/Year slope (95% CrI) |
|---|---|---|---|---|
| | | Age × Frontal | 0.11(−0.05, 0.27) | 0.001(−0.005, 0.007) |
| | | Age × Central | 0.19 (0.04, 0.33) | −0.007 (−0.012, −0.001) |
| Age× Channel | 0.17 | Age × Posterior | 0.12 (−0.18, 0.43) | −0.004(−0.019, 0.012) |
| | | Age × Verbal | 0.10 (−0.01, 0.25) | −0.001 (−0.006, 0.004) |
| | | Age × Spatial | 0.16 (−0.21, 0.56) | −0.010 (−0.022, 0.002) |
| | | Age × Emotional | 0.22 (−0.33, 0.71) | 0.010 (−0.048, 0.028) |
| Age× Task | 0.06 | Age × Motor | 0.27 (−0.14, 0.68) | 0.009 (−0.028, 0.010) |
| | | Frontal × Fast SP | 0.18 (0.06, 0.29) | |
| | | Central × Fast SP | 0.06 (−0.05,0.16) | |
| | | Posterior × Fast SP | −0.01 (−0.18, 0.16) | |
| | | Frontal × Slow SP | 0.02 (−0.11, 0.14) | |
| | | Central × Slow SP | 0.04 (−0.09, 0.17) | |
| Channel× Spindle | 0.16 | Posterior × Slow SP | −0.05 (−0.23, 0.44) | |
| Time-lag | 0.00 | Time-lag bias | | −0.010 (−0.050, 0.026) |
| | 0.02 | N(0, 2.5), InvGamma(2, 0.5) | 0.07 (−0.01, 0.14) | |
| Prior sensitivity | 1.00 | Non-informative | 0.07 (0.01, 0.13) | |
| Main model | 1 − Weight | None | 0.07 (0.01, 0.13) | |

**Appendix 3—table 2.** Summary of interaction and sensitivity models for the SP amplitude–memory association.

| Models | Weight | Factors | Estimate (95% CrI) | Age/Year slope (95% CrI) |
|---|---|---|---|---|
| | | Age × Frontal | 0.17 (−0.02, 0.36) | −0.002 (−0.008, 0.003) |
| | | Age × Central | 0.15 (−0.05, 0.34) | −0.002 (−0.009, 0.004) |
| Age× Channel | 0.00 | Age × Posterior | 0.13 (−0.28, 0.55) | −0.008 (−0.032, 0.016) |

*Appendix 3—table 2 Continued on next page*

*Appendix 3—table 2 Continued*

| Models | Weight | Factors | Estimate (95% CrI) | Age/Year slope (95% CrI) |
|---|---|---|---|---|
| | | Age × Verbal | 0.10 (−0.09, 0.31) | 0.001 (−0.005, 0.007) |
| | | Age × Spatial | 0.38 (0.08, 0.67) | −0.009 (−0.016, −0.001) |
| | | Age × Emotional | 0.20 (−0.46, 0.73) | −0.015 (−0.058, 0.028) |
| Age× Task | 0.73 | Age × Motor | −0.26 (−0.80, 0.35) | 0.013 (−0.018, 0.044) |
| | | Frontal × Fast SP | 0.14 (-0.03, 0.32) | |
| | | Central × Fast SP | 0.13 (−0.03, 0.23) | |
| | | Posterior × Fast SP | 0.05 (−0.17, 0.28) | |
| | | Frontal × Slow SP | 0.11 (−0.07, 0.28) | |
| | | Central × Slow SP | −0.01 (−0.20, 0.19) | |
| Channel× Spindle | 0.00 | Posterior × Slow SP | −0.18 (−0.46, 0.10) | |
| Time-lag | 0.00 | Time-lag bias | | 0.00 (−0.070, 0.070) |
| | 0.00 | $N$(0, 2.5), InvGamma(2, 0.5) | 0.07 (−0.04, 0.18) | |
| Prior sensitivity | 1.00 | Non-informative | 0.07 (−0.04, 0.19) | |
| Main model | 1 − Weight | None | 0.07 (−0.04, 0.18) | |

**Appendix 3—table 3.** Summary of interaction and sensitivity models for the coupling strength–memory association.

| Models | Weight | Factors | Estimate (95% CrI) | Age/Year slope (95% CrI) |
|---|---|---|---|---|
| | | Age × Frontal | 0.14 (−0.03, 0.30) | −0.001 (−0.007, 0.004) |
| | | Age × Central | 0.18 (0.01, 0.33) | −0.005 (−0.011, 0.001) |
| Age× Channel | 0.00 | Age × Posterior | −0.06 (−0.38, 0.27) | 0.009 (−0.009, 0.026) |
| | | Age × Verbal | 0.11 (−0.04, 0.27) | −0.002 (−0.007, 0.003) |
| | | Age × Spatial | 0.13 (−0.18, 0.45) | −0.004 (−0.015, 0.007) |
| | | Age × Emotional | 0.17 (−0.39, 0.64) | −0.001 (-0.035, 0.035) |
| Age× Task | 0.00 | Age × Motor | 0.14 (−0.45, 0.67) | −0.004(−0.028, 0.023) |
| | | Frontal × Fast SP | 0.15 (0.01, 0.28) | |
| | | Central × Fast SP | 0.02 (−0.09, 0.14) | |
| | | Posterior × Fast SP | 0.08 (−0.12, 0.28) | |
| | | Frontal × Slow SP | 0.04 (−0.11, 0.18) | |
| | | Central × Slow SP | 0.05 (0.10, 0.20) | |
| Channel× Spindle | 0.00 | Posterior × Slow SP | 0.15 (−0.05, 0.35) | |
| Time-lag | 0.43 | Time-lag bias | | 0.024 (−0.015, 0.060) |
| | 0.26 | $N$(0, 2.5), InvGamma(2, 0.5) | 0.08 (0.00, 0.16) | |
| Prior sensitivity | 1.00 | Non-informative | 0.08 (0.02, 0.15) | |
| Main model | 1 − Weight | None | 0.08 (0.02, 0.15) | |

**Appendix 3—table 4.** Summary of interaction and sensitivity models for the coupling percentage–memory association.

| Models | Weight | Factors | Estimate (95% CrI) | Age/Year slope (95% CrI) |
|---|---|---|---|---|
| | | Age × Frontal | −0.04 (−0.76, 0.72) | 0.005 (−0.038, 0.045) |
| | | Age × Central | −0.00 (−0.59, 0.63) | -0.001 (-0.036, 0.031) |
| Age× Channel | 0.40 | Age × Posterior | 0.18 (−0.61, 0.98) | −0.014 (−0.062, 0.030) |
| | | Age × Verbal | 0.42 (−0.24, 1.16) | −0.035 (−0.085, 0.008) |
| | | Age × Spatial | −0.04 (−1.00, 1.07) | 0.003 (−0.072, 0.072) |
| | | Age × Emotional | 0.21 (-0.54, 0.84) | −0.017 (−0.073, 0.034) |
| Age× Task | 0.65 | Age × Motor | 0.39 (−0.91, 1.49) | −0.015 (−0.085, 0.055) |
| | | Frontal × Fast SP | 0.02 (−0.16, 0.19) | |
| | | Central × Fast SP | −0.04 (−0.20, 0.13) | |
| | | Posterior × Fast SP | 0.00 (−0.29, 0.27) | |
| | | Frontal × Slow SP | 0.10 (−0.10, 0.29) | |
| | | Central × Slow SP | 0.01 (−0.18, 0.19) | |
| Channel× Spindle | 0.00 | Posterior × Slow SP | −0.07 (−0.36, 0.95) | |
| Time-lag | NaN | Time-lag bias | | *Did not perform* |
| | 0.09 | $N(0, 2.5)$, InvGamma(2, 0.5) | −0.04 (−0.17, 0.09) | |
| Prior sensitivity | 1.00 | Non-informative | −0.03 (−0.16, 0.07) | |
| Main model | 1 − Weight | None | −0.03 (−0.15, 0.07) | |

## Appendix 4

## Publication bias

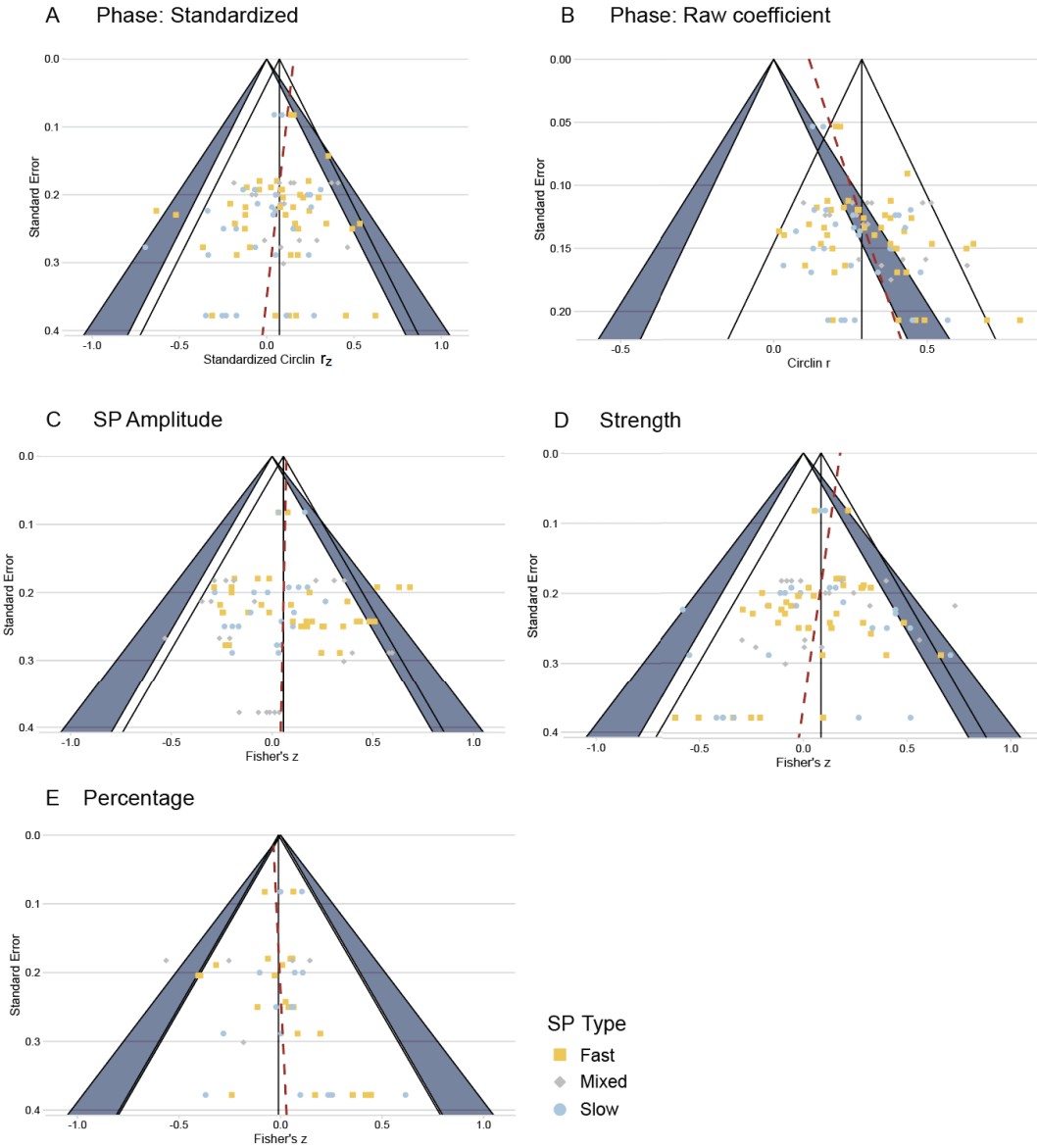

**Appendix 4—figure 1.** Funnel Plot and Egger regression for assessing publication bias. Each colored dot represents one effect size and corresponding standard error. The outer border of the transparent triangle indicates the area where it is anticipated that 95% of the included studies would fall if there were no publication biases present. The red dashed line represents the superimposed Egger's regression line. We noted that with sufficient sample sizes, the phase-memory association and strength-memory association tend towards a moderate magnitude.

# Appendix 5

## Simulation of circular-linear correlation and standardization

From **Appendix 5—figure 1**, we observed that the mean of the raw circular-linear correlation coefficient (Circlin $r$) shifts with the change of sample sizes, and the distribution is highly skewed. By approximating the distribution of the weighted circular-linear distribution to the chi-square distribution with 2 degrees of freedom (**Fisher et al., 1993**), $\chi^2_2$, we derived that the circular-linear correlation coefficient has the following properties in an approximate form:

$$E(\rho) = \sqrt{\frac{\pi}{2n}}, \quad \mathrm{Var}(\rho) = \frac{4 - \pi}{2n}, \quad 0 \leq \rho \leq 1, \ n \geq 2 \tag{1}$$

By standardizing the circular-linear correlation coefficient (see Methods: Standardized circular-linear correlation coefficient), we first generated underlying populations that have null (0), moderate (0.3–0.4), or large (0.6–0.7) $r_z$ correlations, then tested whether the sampling distribution drawn from these populations followed a normal distribution across varying sample sizes.

We can observe from **Appendix 5—figure 1** that under the null hypothesis, the standardized Circlin $r_z$ follows similar distributions with the linear Pearson's $r$. It is centered at 0 and can approximate unbounded normal after Fisher's $z$ transformation. The difference between Pearson's $r$ and standardized Circlin $r_z$ is that the negative x-axis of the Circlin $r_z$ represents a magnitude of correlation under the null hypothesis without effects, instead of a negative direction of the association.

As long as the sample size is more significant than 25, the performance of the Circlin transformation is highly stable. However, we should acknowledge that the non-linear correlation is unreliable under small sample sizes. Therefore, we do not recommend conducting circular-linear correlational analysis when $n < 15$.

To ensure the normality of standardized Circlin $r_z$ under large effect size of correlations, we examined sampling distributions where the underlying population $r_z$ ranged from 0.3 to 0.4 (moderate correlation) and 0.6–0.7 (strong correlation). After Fisher's $z$ transformation, these distributions consistently aligned with superimposed normal distributions (see **Appendix 5—figure 2**), and they performed robustly even when sample sizes were relatively small.

Therefore, we encourage future studies to report the standardized coefficient instead of the raw Circlin $r$, which can accurately reflect the true effect size without exaggeration when the sample size is small, as well as improve the clarity of interpretation and comparability with other types of correlation coefficients across studies.

The code used in R to standardize the circular-linear coefficient is:

```
limma::zscore(n * r^2, dist = 'chisq', df = 2) / sqrt(n)
```

where $n$ represents the sample size while $r$ represents the Circlin coefficient.

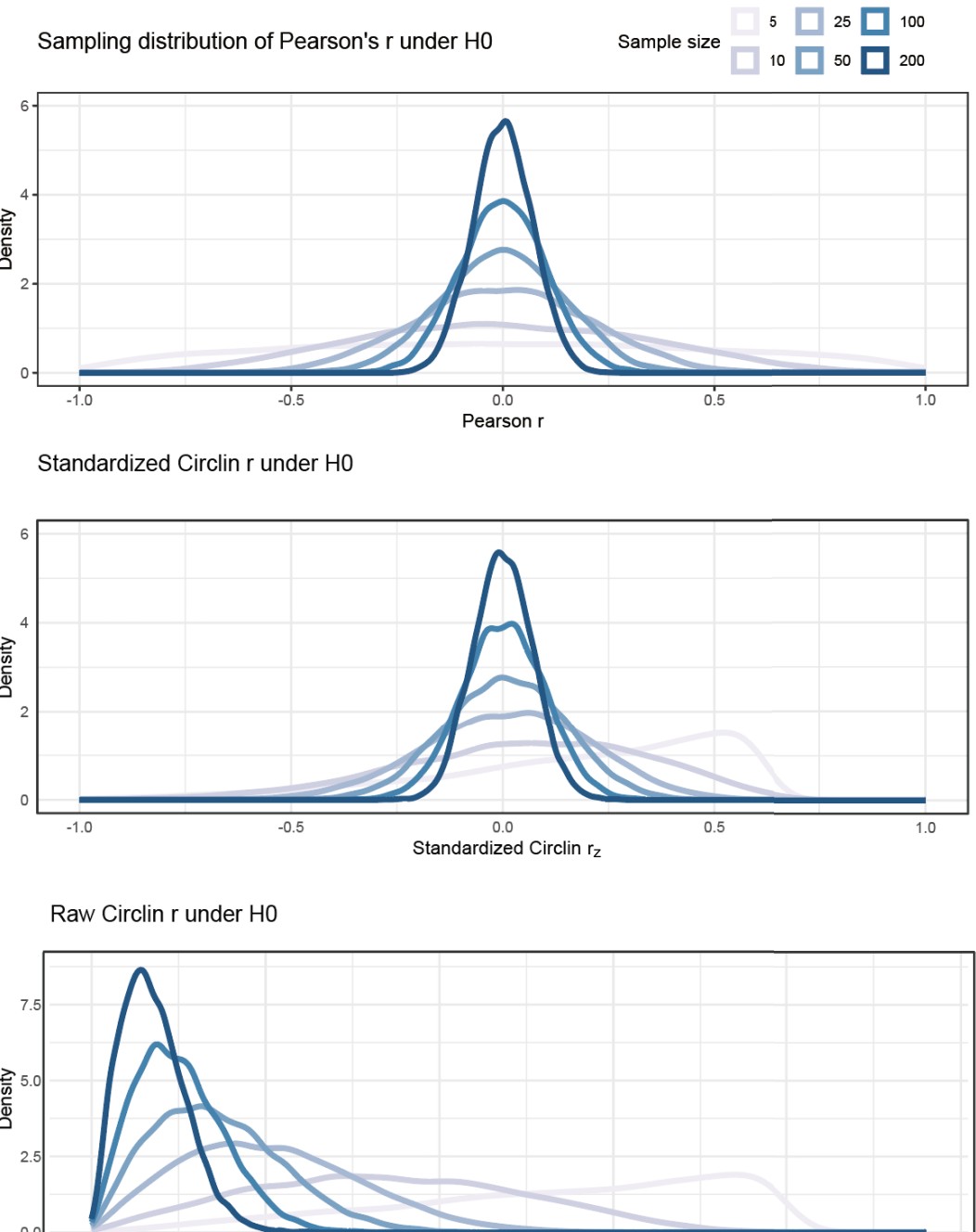

**Appendix 5—figure 1.** Comparison of sampling distributions of standardized Circlin $r_z$ under null.

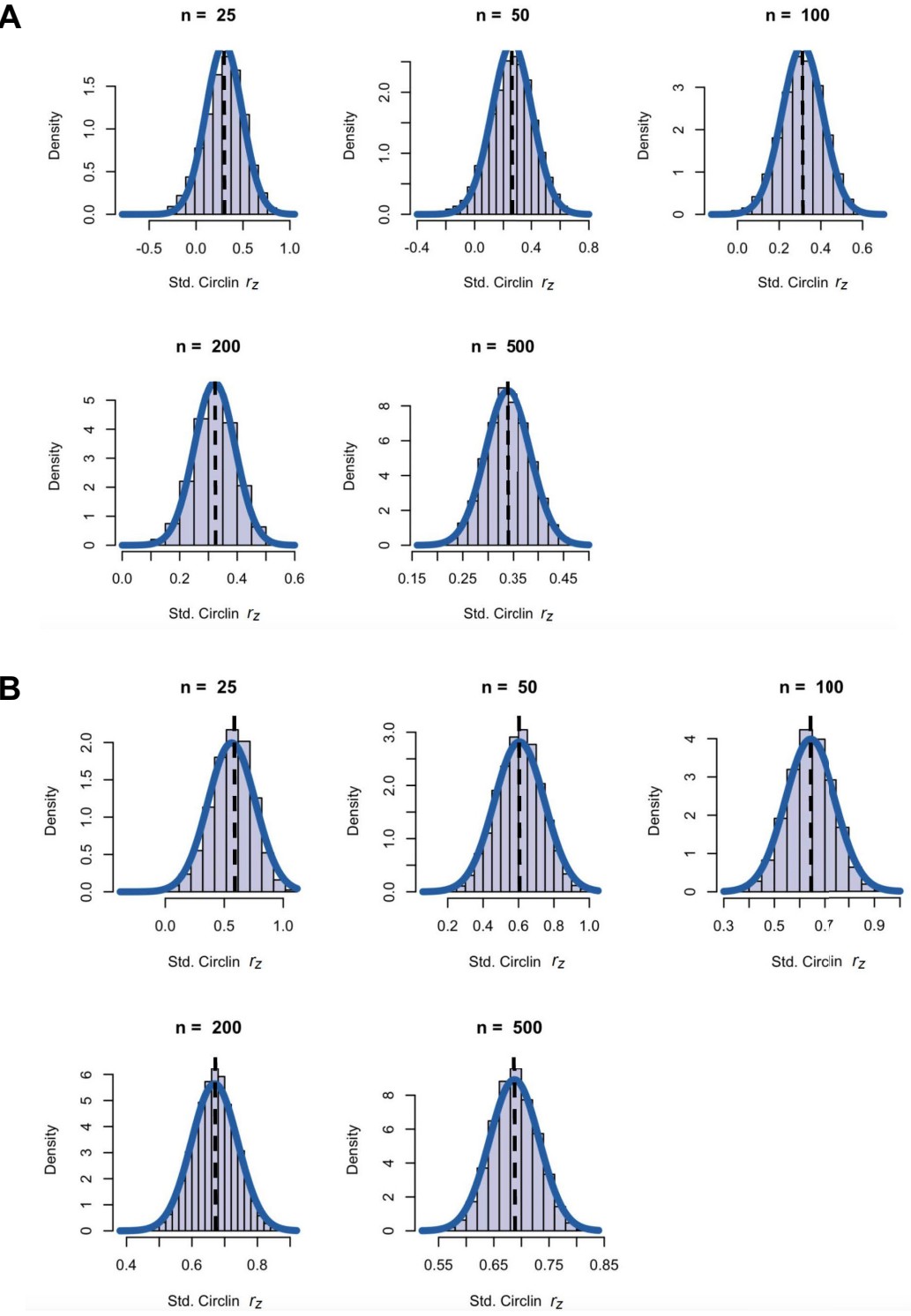

**Appendix 5—figure 2.** Sampling distributions of standardized Circlin $r_z$ drawn from population correlations. (**A**) Sampling distributions of standardized Circlin drawn from populations with moderate correlations ($r_z$ = 0.3–0.4). Vertical dashed lines indicate the true population values from which samples were generated. (**B**) Sampling distributions of standardized Circlin drawn from populations with strong correlations ($r_z$ = 0.6–0.7).

## Appendix 6

### SO-slow SP spatiotemporal analysis

Contrary to fast SPs, the preferred phase of SO-slow SP coupling did not show a significant quadratic association with memory retention in any topographic regions (see *Appendix 6—figure 1A*), all $r \leq 0.1$, $r_z \leq 0.01$, $p > 0.05$. After accounting for repeated measurements, we found that SO-slow SP coupling phase occurs slightly before the down-state trough of SOs, reflected by the phase in frontal (2.71 rad [2.23,–3.14]), central (3.02 rad [2.64,–2.93]) and posterior regions (2.82 rad [2.20,–2.84]). However, the phase distribution across participants is considerably less consistent than SO-fast SP coupling (all $z < 0.34$, $p < 0.01$, Rayleigh test), which may relate to between-study variability in the definition of slow SPs.

No significant phase shift was observed between frontal and central areas, $r = \Delta - 0.31$ rad, $BF_{10} = 0.06$, probability = 0.05; between frontal and posterior areas, $r = \Delta - 0.11$ rad, $BF_{10} = 0.49$, probability = 0.33; or between central and posterior areas, $r = \Delta 0.20$ rad, $BF_{10} = 3.64$, probability = 0.78. In summary, there is no strong evidence supporting an association between the SO-slow SP coupling phase and memory retention performance, or to substantiate phase shifts across cortical areas.

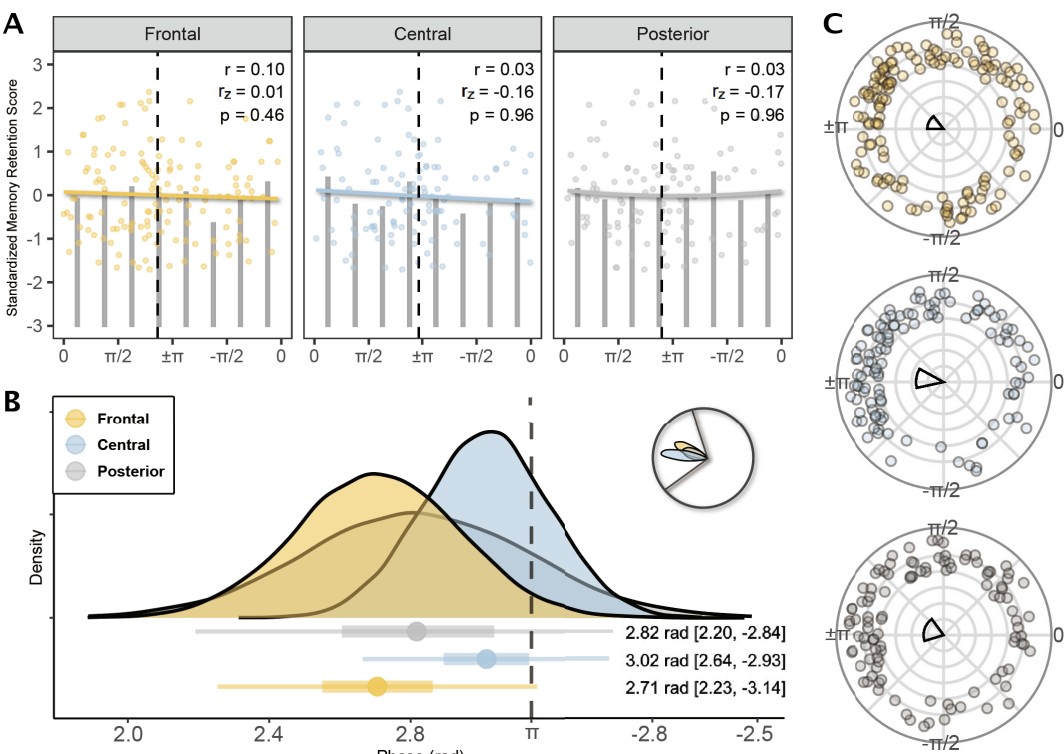

**Appendix 6—figure 1.** Preferred slow oscillation-slow spindle coupling phase and its association with memory retention. (**A**) Quadratic regression of the phase-memory association under different regions of PSG channels aggregated from studies included in the meta-analysis. *0* peak of SO upstate; *±π* trough of SO downstate; *r* circular-linear correlation coefficient; *$r_z$* standardized circular-linear correlation coefficient. Bars represent the mean memory retention scores per *π/4* radian (45°). The dashed vertical line represents the mean preferred phase across studies. The colored quadratic fit line represents the direction of the relationship. None of the PSG channels displays a typical quadratic relationship around the down-state trough of SOs. (**B**) Posterior distributions of mean preferred phases from the Bayesian circular mixed-effect model. The circular posterior distribution is shown in the top right corner, and the area between two black lines is projected on a linear scale in the main graph. The vertical line reflects the down-state trough of SOs. Dots and error bars denote the mean and 95% credible intervals of phases detected from each channel cluster. Phase values are reported in radians. (**C**) Circular plot of the preferred coupling phase. From top to bottom, frontal, central, and posterior. The direction of each colored dot represents the preferred coupling phase of each subject recorded from PSG channels in each cluster. The direction of the mean resultant vector indicates the mean preferred coupling phase across subjects, the width indicates the 95% credible interval of the mean coupling phase, the length from 0 (center) to 1 (circumference) indicates the consistency of coupling phase across subjects.

