## [Editor Report · eLife Assessment]

This **important** study presents a meta-analysis confirming a statistically significant association between slow oscillation-spindle coupling and memory formation, although the reported effects are limited (~0.5% of variance). The evidence is overall **convincing**, but the statistical methods may be difficult to follow for readers unfamiliar with advanced techniques. This work will be of particular interest to neuroscientists studying the neural mechanisms of sleep and memory.

---

## [Referee Report · Reviewer #1 (Public review)]

In this meta-analysis, Ng and colleagues review the association between slow-oscillation spindle coupling during sleep and overnight memory consolidation. The coupling of these oscillations (and also hippocampal sharp-wave ripples) have been central to theories and mechanistic models of active systems consolidation, that posit that the coupling between ripples, spindles, and slow oscillations (SOs) coordinate and drive the coordinated reactivation of memories in hippocampus and cortex, facilitating cross-regional information and ultimately memory strengthening and stabilisation.

Given the importance that these coupling mechanisms have been given in theory, this is a timely and important contribution to the literature in terms of determining whether these theoretical assumptions hold true in human data. The results show that the timing of sleep spindles relative to the SO phase, and the consistency of that timing, predicted overnight memory consolidation in meta-analytic models. The overall amount of coupling events did not show as strong a relationship. Coupling phase in particular was moderated by a number of variables including spindle type (fast, slow), channel location (frontal, central, posterior), age, and memory type. The main takeaway is that fast spindles that consistently couple close to the peak of the SO in frontal channel locations are optimal for memory consolidation, in line with theoretical predictions. These findings will be very useful for future researchers in terms of determining necessary sample sizes to observe coupling - memory relationships, and in the selection and reporting of relevant coupling metrics.

Although the meta-analysis covers the three main coupling metrics that are typically assessed (occurrence, timing, and consistency), the meta-analysis also includes spindle amplitude. This may be confusing to readers, as this is not a measurement of SO-spindle coupling but instead a measurement of spindles in general (which may or may not be coupled).

---

## [Referee Report · Reviewer #2 (Public review)]

This article reviews the studies on the relationship between slow oscillation (SO)-spindle (SP) coupling and memory consolidation. It innovatively employs non-normal circular linear correlations through a Bayesian meta-analysis. A systematic analysis of the retrieved studies highlighted that co-coupling of SO and the fast SP's phase and amplitude at the frontal part better predicts memory consolidation performance.

Regarding the moderator of age, this study not only provided evidence of the effect across all age groups but also the effect in a younger age group (without the small sample of elders that has a large gap from the younger age groups). The ageing effects become less pronounced, but the model still shows a moderate effect.

---

## [Referee Report · Reviewer #3 (Public review)]

This manuscript presents a meta-analysis of 23 studies, which report 297 effect sizes, on the effect of SO-spindle coupling on memory performance. The analysis has been done with great care, and the results are described in great detail. In particular, there are separate analyses for coupling phase, spindle amplitude, coupling strength (e.g., measured by vector length or modulation index), and coupling percentage (i.e., the percentage of SPs coupled with SOs). The authors conclude that the precision and strength of coupling showed significant correlations with memory retention.

There are two main points where I do not agree with the authors.

First, the authors conclude that "SO-SP coupling should be considered as a general physiological mechanism for memory consolidation". However, the reported effect sizes are smaller than what is typically considered a "small effect" (0.10)

Second, the study implements state-of-the-art Bayesian statistics. While some might see this as a strength, I would argue that it is not. A classical meta-analysis is relatively easy to understand, even for readers with only a limited background in statistics. A Bayesian analysis, on the other hand, introduces a number of subjective choices that render it much less transparent. This becomes obvious in the forest plots. It is not immediately apparent to the reader how the distributions for each study represent the reported effect sizes (gray dots), which makes the analyses unnecessarily opaque. It is commendable that the authors now provide classical forest plots as Figs. S10.1-4.

However, analyses that require a "Markov chain Monte Carlo (MCMC) method, [..] with the no-U-turn Hamiltonian Monte Carlo (HMC) samplers, [..] with each chain undergoing 12,000 iterations (including 2,000 warm-ups)" for calculating accurate Bayes Factors (BF), and checking its convergence "through graphical posterior predictive checks, [..] trace plots, and [..] Gelman and Rubin Diagnostic", which should then result in something resembling "a uniformly undulating wave with high overlap between chains" still seems overly complex. It follows a recent trend in using more and more opaque methods. Where we had to trust published results a decade ago because the data were not openly available, today we must trust the results because methods (including open source software toolboxes) can no longer be checked with reasonable effort.

---

## [Author Response]

The following is the authors’ response to the original reviews.

**Reviewer #1 (Public review):**
Given the importance that these coupling mechanisms have been given in theory, this is a timely and important contribution to the literature in terms of determining whether these theoretical assumptions hold true in human data.

Thank you!

I did not follow the logic behind including spindle amplitude in the meta-analysis. This is not a measure of SO-spindle coupling (which is the focus of the review), unless the authors were restricting their analysis of the amplitude of coupled spindles only. It doesn't sound like this is the case though. The effect of spindle amplitude on memory consolidation has been reviewed in another recent meta-analysis (Kumral et al, 2023, Neuropsychologia). As this isn't a measure of coupling, it wasn't clear why this measure was included in the present meta-analysis. You could easily make the argument that other spindle measures (e.g., density, oscillatory frequency) could also have been included, but that seems to take away from the overall goal of the paper which was to assess coupling.

Indeed, spindle amplitude refers to all spindle events rather than only coupled spindles. This choice was made because we recognized the challenge of obtaining relevant data from each study—only 4 out of the 23 included studies performed their analyses after separating coupled and uncoupled spindles. This inconsistency strengthens the urgency and importance of this meta-analysis to standardize the methods and measures used for future analysis on SO-SP coupling and beyond. We agree that focusing on the amplitude of coupled spindles would better reveal their relations with coupling, and we have discussed this limitation in the manuscript.

Nevertheless, we believe including spindle amplitude in our study remains valuable, as it served several purposes. First, SO-SP coupling involves the modulation between spindle amplitude and slow oscillation phase. Different studies have reported conflicting conclusions regarding how overall spindle amplitude was related to coupling as an indicator of oscillation strength overnight– some found significant correlations (e.g., Baena et al., 2023), while others did not (e.g., Roebber et al., 2022). This discrepancy highlights an indirect but potentially crucial insight into the role of spindle amplitude in coupling dynamics. Second, in studies related to SO-SP coupling, spindle amplitude is one of the most frequently reported measures along with other coupling measures that significantly correlated with oversleep memory improvements (e.g. Kurz et al., 2023; Ladenbauer et al., 2021; Niknazar et al., 2015), so we believe that including this measure can provide a more comprehensively review of the existing literature on SO-SP coupling. Third, incorporating spindle amplitude allows for a direct comparison between the measurement of coupling and individual events alone in their contribution to memory consolidation– a question that has been extensively explored in recent research. (e.g., Hahn et al., 2020; Helfrich et al., 2019; Niethard et al., 2018; Weiner et al., 2023). Finally, spindle amplitude was identified as the most important moderator for memory consolidation in Kumral et al.'s (2023) meta-analysis. By including it in our analysis, we sought to replicate their findings within a broader framework and introduce conceptual overlaps with existing reviews. Therefore, although we were not able to selectively include coupled spindles, there is still a unique relation between spindle amplitude and SO-SP coupling that other spindle measures do not have.

Originally, we also intended to include coupling density or counts in the analysis, which seems more relevant to the coupling metrics. However, the lack of uniformity in methods used to measure coupling density posed a significant limitation. We hope that our study will encourage consistent reporting of all relevant parameters in future research, allowing future meta-analyses to incorporate these measures comprehensively. We have added this discussion to the revised version of the manuscript (*p*. 3) to further clarify these points.

All other citations were referenced in the manuscript.

At the end of the first paragraph of section 3.1 (page 13), the authors suggest their results "... further emphasise the role of coupling compared to isolated oscillation events in memory consolidation". This had me wondering how many studies actually test this. For example, in a hierarchical regression model, would coupled spindles explain significantly more variance than uncoupled spindles? We already know that spindle activity, independent of whether they are coupled or not, predicts memory consolidation (e.g., Kumral meta-analysis). Is the variance in overnight memory consolidation fully explained by just the coupled events? If both overall spindle density and coupling measures show an equal association with consolidation, then we couldn't conclude that coupling compared to isolated events is more important.

While primary coupling measurements, including coupling phase and strength, showed strong evidence for their associations with memory consolidation, measures of spindles, including spindle amplitude, only exhibited limited evidence (or “non-significant” effect) for their association with consolidation. These results are consistent with multiple empirical studies using different techniques (e.g., Hahn et al., 2020; Helfrich et al., 2019; Niethard et al., 2018; Weiner et al., 2023), which reported that coupling metrics are more robust predictors of consolidation and synaptic plasticity than spindle or slow oscillation metrics alone. However, we agree with the reviewer that we did not directly separate the effect between coupled and uncoupled spindles, and a more precise comparison would involve contrasting the “coupling of oscillation events” with ”individual oscillation events” rather than coupling versus isolated events.

We recognized that Kumral and colleagues’ meta-analysis reported a moderate association between spindle measures and memory consolidation (e.g., for spindle amplitude-memory association they reported an effect size of approximately *r* = 0.30). However, one of the advantages of our study is that we actively cooperated with the authors to obtain a large number of unreported and insignificant data relevant to our analysis, as well as separated data that were originally reported under mixed conditions. This approach decreases the risk of false positives and selective reporting of results, making the effect size more likely to approach the true value. In contrast, we found only a weak effect size of *r* = 0.07 with minimal evidence for spindle amplitude-memory relation. However, we agree with the reviewer that using a more conservative term in this context would be a better choice since we did not measure all relevant spindle metrics including the density.

To improve clarity in our manuscript, we have revised the statement to: “Together with other studies included in the review, our results suggest a crucial role of coupling but did not support the role of spindle events alone in memory consolidation,” and provide relevant references (*p*. 13). We believe this can more accurately reflect our findings and the existing literature to address the reviewer’s concern.

It was very interesting to see that the relationship between the fast spindle coupling phase and overnight consolidation was strongest in the frontal electrodes. Given this, I wonder why memory promoting fast spindles shows a centro-parietal topography? Surely it would be more adaptive for fast spindles to be maximally expressed in frontal sites. Would a participant who shows a more frontal topography of fast spindles have better overnight consolidation than someone with a more canonical centro-parietal topography? Similarly, slow spindles would then be perfectly suited for memory consolidation given their frontal distribution, yet they seem less important for memory.

Regarding the topography of fast spindles and their relationship to memory consolidation, we agree this is an intriguing issue, and we have already developed significant progress in this topic in our ongoing work, and have found evidence that participants with a more frontal topography of fast spindles show better overnight consolidation. These findings will be presented in our future publications. We share a few relevant observations: First, there are significant discrepancies in the definition of “slow spindle” in the field. Some studies defined slow spindle from 9-12 Hz (e.g. Mölle et al., 2011; Kurz et al., 2021), while others performed the event detection within a range of 11-13/14 Hz and found a frontal-dominated topography (e.g. Barakat et al., 2011; D'Atri et al., 2018). Compounding this issue, individual and age differences in spindle frequency are often overlooked, leading to challenges in reliably distinguishing between slow and fast spindles. Some studies have reported difficulty in clearly separating the two types of spindles altogether (e.g., Hahn et al., 2020). Moreover, a critical factor often ignored in past research is the propagating nature of both slow oscillations and spindles across the cortex, where spindles are coupled with significantly different phases of slow oscillations (see Figure 5). In addition, the frontal region has the strongest and most active SOs as its origin site, which may contribute to the role of frontal coupling. In contrast, not all SOs propagate from PFC to centro-parietal sites. The reviewer also raised an interesting idea that slow spindles would be perfectly suited for memory consolidation given their frontal distribution. We propose that one possible explanation is that if SOs couple exclusively with slow SPs, they may lose their ability to coordinate inter-area activity between centro-parietal and frontal regions, which could play a critical role in long-range memory transmission across hippocampus, thalamus, and prefrontal cortex. This hypothesis requires investigation in future studies. We believe a better understanding of coupling in the context of the propagation of these waves will help us better understand the observed frontal relationship with consolidation. Therefore, we believe this result supports our conclusion that coupling precision is more important than intensity, and we have addressed this in revised manuscript (*pp*. 15-16).

The authors rightly note the issues with multiple comparisons in sleep physiology and memory studies. Multiple comparison issues arise in two ways in this literature. First are comparisons across multiple electrodes (many studies now use high-density systems with 64+ channels). Second are multiple comparisons across different outcome variables (at least 3 ways to quantify) coupling (phase, consistency, occurrence) x 2 spindle types (fast, slow). Can the authors make some recommendations here in terms of how to move the field forward, as this issue has been raised numerous times before (e.g., Mantua 2018, Sleep; Cox & Fell 2020, Sleep Medicine Reviews for just a couple of examples). Should researchers just be focusing on the coupling phase? Or should researchers always report all three metrics of coupling, and correct for multiple comparisons? I think the use of pre-registration would be beneficial here, and perhaps could be noted by the authors in the final paragraph of section 3.5, where they discuss open research practices.

There are indeed multiple methods that we can discuss, including cluster-based and non-parametric methods, etc., to correct for multiple comparisons in EEG data with spatiotemporal structures. In addition, encouraging the reporting of all tested but insignificant results, at least in supplementary materials, is an important practice that helps readers understand the findings with reduced bias. We agree with the reviewer’s suggestions and have added more information in section 3.4-3.5 (*p*. 17) to advocate for a standardized “template” used to report effect sizes and correct multiple comparisions in future research.

We advocate for the standardization of reporting all three coupling metrics– phase, strength, and prevalence (density, count, and/or percentage coupled). Each coupling metric captures distinct a property of the coupling process and may interact with one another (Weiner et al., 2023). Therefore, we believe it is essential to report all three metrics to comprehensively explore their different roles in the “how, what, and where” of long-distance communication and consolidation of memory. As we advance toward a deeper understanding of the relationship between memory and sleep, we hope this work establishes a standard for the standardization, transparency, and replication of relevant studies.

**Reviewer #2 (Public review):**
Regarding the Moderator of Age: Although the authors discuss the limited studies on the analysis of children and elders regarding age as a moderator, the figure shows a significant gap between the ages of 40 and 60. Furthermore, there are only a few studies involving participants over the age of 60. Given the wide distribution of effect sizes from studies with participants younger than 40, did the authors test whether removing studies involving participants over 60 would still reveal a moderator effect?

We agree that there is an age gap between younger and older adults, as current studies often focus on contrasting newly matured and fully aged populations to amplify the effect, while neglecting the gradual changes in memory consolidation mechanisms across the aging spectrum. We suggest that a non-linear analysis of age effects would be highly valuable, particularly when additional child and older adult data become available.

In response to the reviewer’s suggestion, we re-tested the moderation effect of age after excluding effect sizes from older adults. The results revealed a decrease in the strength of evidence for phase-memory association due to increased variability, but were consistent for all other coupling parameters. The mean estimations also remained consistent (coupling phase-memory relation: -0.005 [-0.013, 0.004], *BF*10 = 5.51, the strength of evidence reduced from strong to moderate; coupling strength-memory relation: -0.005 [-0.015, 0.008], *BF*10 = 4.05, the strength of evidence remained moderate). These findings align with prior research, which typically observed a weak coupling-memory relationship in older adults during aging (Ladenbauer et al, 2021; Weiner et al., 2023) but not during development (Hahn et al., 2020; Kurz et al., 2021; Kurz et al., 2023). Therefore, this result is not surprising to us, and there are still observable moderate patterns in the data. We have reported these additional results in the revised manuscript (pp. 6, 11), and interpret “the moderator effect of age in the phase-memory association becomes less pronounced during development after excluding the older adult data”. We believe the original findings including the older adult group remain meaningful after cautious interpretation, given that the older adult data were derived from multiple studies and different groups, and they represent the aging effects.

**Reviewer #3 (Public review):**
First, the authors conclude that "SO-SP coupling should be considered as a general physiological mechanism for memory consolidation". However, the reported effect sizes are smaller than what is typically considered a "small effect”.

While we acknowledge the concern about the small effect sizes reported in our study, it is important to contextualize these findings within the field of neuroscience, particularly memory research. Even in individual studies, small effect sizes are not uncommon due to the inherent complexity of the mechanisms involved and the multitude of confounding variables. This is an important factor to be considered in meta-analyses where we synthesize data from diverse populations and experimental conditions. For example, the relationship between SO-slow SP coupling and memory consolidation in older adults is expected to be insignificant.

As Funder and Ozer (2019) concluded in their highly cited paper, an effect size of *r* = 0.3 in psychological and related fields should be considered large, with *r* = 0.4 or greater likely representing an overestimation and rarely found in a large sample or a replication. Therefore, we believe *r* = 0.1 should not be considered as a lower bound of the small effect. Bakker et al. (2019) also advocate for a contextual interpretation of the effect size. This is particularly important in meta-analyses, where the results are less prone to overestimation compared to individual studies, and we cooperated with all authors to include a large number of unreported and insignificant results. In this context, small correlations may contain substantial meaningful information to interpret. Although we agree that effect sizes reported in our study are indeed small at the overall level, they reflect a rigorous analysis that incorporates robust evidence across different levels of moderators. Our moderator analyses underscore the dynamic nature of coupling-memory relationships, with stronger associations observed in moderator subgroups that have historically exhibited better memory performance, particularly after excluding slow spindles and older adults. For example, both the coupling phase and strength of frontal fast spindles with slow oscillations exhibited "moderate-to-large" correlations with the consolidation of different types of memory, especially in young adults, with *r* values ranging from 0.18 to 0.32. (see Table S9.1-9.4). We have included discussion about the influence of moderators and hierarchical structures on the dynamics of coupling-memory associations (*pp*. 17, 20). In addition, we have updated the conclusion to be “SO-**fast** SP coupling should be considered as a general physiological mechanism for memory consolidation” (*p*. 1).

Second, the study implements state-of-the-art Bayesian statistics. While some might see this as a strength, I would argue that it is the greatest weakness of the manuscript. A classical meta-analysis is relatively easy to understand, even for readers with only a limited background in statistics. A Bayesian analysis, on the other hand, introduces a number of subjective choices that render it much less transparent.This kind of analysis seems not to be made to be intelligible to the average reader. It follows a recent trend of using more and more opaque methods. Where we had to trust published results a decade ago because the data were not openly available, today we must trust the results because the methods can no longer be understood with reasonable effort.This becomes obvious in the forest plots. It is not immediately apparent to the reader how the distributions for each study represent the reported effect sizes (gray dots). Presumably, they depend on the Bayesian priors used for the analysis. The use of these priors makes the analyses unnecessarily opaque, eventually leading the reader to question how much of the findings depend on subjective analysis choices (which might be answered by an additional analysis in the supplementary information).

We appreciate the reviewer for sharing this viewpoint and we value the opportunity to clarify some key points. To address the concern about clarity, we have included more details in the methods section explaining how to interpret Bayesian statistics including priors, posteriors, and Bayes factors, making our results more accessible to those less familiar with this approach.

On the use of Bayesian models, we believe there may have been a misunderstanding. Bayesian methods, far from being "opaque" or overly complex, are increasingly valued for their ability to provide nuanced, accurate, and transparent inferences (Sutton & Abrams, 2001; Hackenberger, 2020; van de Schoot et al., 2021; Smith et al., 1995; Kruschke & Liddell, 2018). It has been applied in more than 1,200 meta-analyses as of 2020 (Hackenberger, 2020). In our study, we used priors that assume no effect (mean set to 0, which aligns with the null) while allowing for a wide range of variation to account for large uncertainties. This approach reduces the risk of overestimation or false positives and demonstrates much-improved performance over traditional methods in handling variability (Williams et al., 2018; Kruschke & Liddell, 2018). In addition, priors can also increase transparency, since all assumptions are formally encoded and open to critique or sensitivity analysis. In contrast, frequentist methods often rely on hidden or implicit assumptions such as homogeneity of variance, fixed-effects models, and independence of observations that are not directly testable. Sensitivity analyses reported in the supplemental material (Table S9.1-9.4) confirmed the robustness of our choices of priors– our results did not vary by setting different priors.

As Kruschke and Liddell (2018) described, “shrinkage (pulling extreme estimates closer to group averages) helps prevent false alarms caused by random conspiracies of rogue outlying data,” a well-known advantage of Bayesian over traditional approaches. This explains the observed differences between the distributions and grey dots in the forest plots, which is an advantage of Bayesian models in handling heterogeneity. Unlike *p*-values, which can be overestimated with a large sample size and underestimated with a small sample size, Bayesian methods make assumptions explicit, enabling others to challenge or refine them– an approach aligned with open science principles (van de Schoot et al., 2021). For example, a credible interval in Bayesian model can be interpreted as “there is a 95% probability that the parameter lies within the interval.”, while a confidence interval in frequentist model means “In repeated experiments, 95% of the confidence intervals will contain the true value.” We believe the former is much more straightforward and convincing for readers to interpret. We will ensure our justification for using Bayesian models is more clearly presented in the manuscript (*pp*. 21-23).

We acknowledge that even with these justifications, different researchers may still have discrepancies in their preferences for Bayesian and frequentist models. To increase the effort of transparent reporting, we have also reported the traditional frequentist meta-analysis results in Supplemental Material 10 to justify the robustness of our analysis, which suggested non-significant differences between Bayesian and frequentist models. We have included clearer references in the updated version of the manuscript to direct readers to the figures that report the statistics provided by traditional models.

However, most of the methods are not described in sufficient detail for the reader to understand the proceedings. It might be evident for an expert in Bayesian statistics what a "prior sensitivity test" and a "posterior predictive check" are, but I suppose most readers would wish for a more detailed description. However, using a "Markov chain Monte Carlo (MCMC) method with the no-U-turn Hamiltonian Monte Carlo (HMC) sampler" and checking its convergence "through graphical posterior predictive checks, trace plots, and the Gelman and Rubin Diagnostic", which should then result in something resembling "a uniformly undulating wave with high overlap between chains" is surely something only rocket scientists understand. Whether this was done correctly in the present study cannot be ascertained because it is only mentioned in the methods and no corresponding results are provided.

We appreciate the reviewer’s concerns about accessibility and potential complexity in our descriptions of Bayesian methods. Our decision to provide a detailed account serves to enhance transparency and guide readers interested in replicating our study. We acknowledge that some terms may initially seem overwhelming. These steps, such as checking the MCMC chain convergence and robustness checks, are standard practices in Bayesian research and are analogous to “linearity”, “normality” and “equal variance” checks in frequentist analysis. In addition, Hamiltonian Monte Carlo (HMC) is the default algorithm Stan (the software we used to fit Bayesian models) uses to sample from the posterior distribution in Bayesian models. It is a type of MCMC method designed to be faster and more efficient than traditional sampling algorithms, especially for complex or high-dimensional models. We have added exemplary plots in the supplemental material S4.1-4.3 and the method section (*pp*. 21-22) to explain the results and interpretation of these convergence checks. We hope this will help address any concerns about methodological rigor.

In one point the method might not be sufficiently justified. The method used to transform circular-linear r (actually, all references cited by the authors for circular statistics use r² because there can be no negative values) into "Z_r", seems partially plausible and might be correct under the H0. However, Figure 12.3 seems to show that under the alternative Hypothesis H1, the assumptions are not accurate (peak Z_r=~0.70 for r=0.65). I am therefore, based on the presented evidence, unsure whether this transformation is valid. Also, saying that Z_r=-1 represents the null hypothesis and Z_r=1 the alternative hypothesis can be misinterpreted, since Z_r=0 also represents the null hypothesis and is not half way between H0 and H1.

First, we realized that in the title of Figures 12.2 and 12.3. “true r = 0.35” and “true r = 0.65” should be corrected as “true r_z” (note that we use r_z instead of Z_r in the revised manuscript per your suggestion). The method we used here is to first generate an underlying population that has null (0), moderate (0.35), or large (0.65) r_z correlations, then test whether the sampling distribution drawn from these populations followed a normal distribution across varying sample sizes. Nevertheless, the reviewer correctly noticed discrepancies between the reported true r_z and its sampling distribution peak. This discrepancy arises because, when generating large population data, achieving exact values close to a strong correlation like r_z = 0.65 is unlikely. We loop through simulations to generate population data and ensure their r_z values fall within a threshold. For moderate effect sizes (e.g., r_z = 0.35), this is straightforward using a narrow range (0.34 < r_z < 0.35). However, for larger effect sizes like r_z = 0.65, a wider range (0.6 < r_z < 0.7) is required. therefore sometimes the population we used to draw the sample has a r_z slightly deviated from 0.65. This remains reasonable since the main point of this analysis is to ensure that a large r_z still has a normal sampling distribution, but not focus specifically on achieving r_z = 0.65.

We acknowledge that this variability of the range used was not clearly explained in supplemental material 12 and it is not accurate to report “true r_z = 0.65”. In the revised version, we have addressed this issue by adding vertical lines to each subplot to indicate the r_z of the population we used to draw samples, making it easier to check if it aligns with the sampling peak. In addition, we have revised the title to “Sampling distributions of r_z drawn from strong correlations

(r_z = 0.6-0.7)”. We confirmed that population r_z and the peak of their sampling distribution remain consistent under both H0 and H1 in all sample sizes with n > 25, and we hope this explanation can fully resolve your concern.

We agree with the reviewer that claiming r_z = -1 represents the null hypothesis is not accurate. The circlin r_z = 0 is better analogous to Pearson’s r = 0 since both represent the mean drawn from the population under the null hypothesis. In contrast, the mean effect size under null will be positive in the raw circlin r, which is one of the important reasons for the transformation. To provide a more accurate interpretation, we updated Table 6 to describe the following strength levels of evidence: no effect (r < 0), null (r = 0), small (r = 0.1), moderate (r = 0.3), and large (r = 0.5). We thank the reviewer again for their valuable feedback.

**Reviewer #2 (Recommendations for the authors):**
(1) There is an extra space in the Notes of Figure 1. "SW R sharp-wave ripple.".

We thank the reviewer for pointing this out. We have confirmed that the "extra space" is not an actual error but a result of how italicized Times New Roman font is rendered in the LaTeX format. We believe that the journal’s formatting process will resolve this issue.

(2) In the introduction, slow oscillations (SO) are defined with a frequency of 0.16-4 Hz, sleep spindles (SP) at 8-16 Hz, and sharp-wave ripples (SWR) at 80-300 Hz. The term "fast oscillation" (FO) is first introduced with the clarification "SPs in our case." However, on page 2, the authors state, "SO-FO coupling involving SWRs, SPs, and SOs..." There seems to be a discrepancy in the definition of FO; does it consistently refer to SPs and SWRs throughout the article?

We appreciate the reviewer’s observation regarding the potential ambiguity of the term "FO." In our manuscript, "FO" is used as a general term to describe the interaction of a "relatively faster oscillation" with a "relatively slower oscillation" in the phase-amplitude coupling mechanism, therefore it is not intended to exclusively refer to SPs or SWRs. For example, it is usually used to describe SO–SP–SWR couplings during sleep memory studies, but Theta–Alpha–Gamma couplings in wakeful memory studies. To address this confusion, we removed the phrase "SPs in our case" and explicitly use "SPs" when referring to spindles. In addition, we have replaced "fast oscillation" with "faster oscillation" to emphasize that it is used in a relative sense (*p*. 1), rather than to refer to a specific oscillation. Also, we only retained the term “FO” when introducing the PAC mechanism.

(3) On page 2, the first paragraph contains the phrase: "...which occur in the precise hierarchical temporal structure of SO-FO coupling involving SWRs, SPs, and SOs ..." Since "SO-FO" refers to slow and fast oscillations, it is better to maintain the order of frequencies, suggesting it as: SOs, SPs, and SWRs.

We sincerely thank the reviewer for their valuable suggestion. We have updated the sentence to maintain the correct order from the lowest to the highest frequencies in the revised version (*p*. 2).

(4) References should be provided:a “Studies using calcium imaging after SP stimulation explained the significance of the precise coupling phase for synaptic plasticity.".b. "Electrophysiology evidence indicates that the association between memory consolidation and SO-SP coupling is influenced by a variety of behavioral and physiological factors under different conditions."c. "Since some studies found that fast SPs predominate in the centroparietal region, while slow SPs are more common in the frontal region, a significant amount of studies only extracted specific types of SPs from limited electrodes. Some studies even averaged all electrodes to estimate coupling..."

This is a great point. These have been referenced as follows:

a. Rephrased: “Studies using calcium imaging and SP stimulation explained the significance of the precise coupling phase for synaptic plasticity.” We changed “after” to “and” to reflect that these were conducted as two separate experiments. This is a summary statement, with relevant citations provided in the following two sentences of the paragraph, including Niethard et al., 2018, and Rosanova et al., 2005. (*p*. 2)

b. Included diverse sources of evidence: “Electrophysiology evidence from studies included in our meta-analysis (e.g. Denis et al., 2021; Hahn et al., 2020; Mylonas et al., 2020) and others (e.g. Bartsch et al., 2019; Muehlroth et al., 2019; Rodheim et al., 2023) reported that the association between memory consolidation and SO-SP coupling is influenced by a variety of behavioral and physiological factors under different conditions.” (*p*. 3)

c. Added references and more details: “Since some studies found that fast SPs predominate in the centroparietal region, while slow SPs are more common in the frontal region, a significant amount of studies selectively extracted specific types of SPs from limited electrodes (e.g. Dehnavi et al., 2021; Perrault et al., 2019; Schreiner et al., 2021). Some studies even averaged all electrodes in their spectral and/or time-series analysis to estimate metrics of oscillations and their couplings (e.g. Denis et al., 2022; Mölle et al., 2011; Nicolas et al., 2022).” (*p*. 4)

**Reviewer #3 (Recommendations for the authors):**

There are a number of terms that are not clearly defined or used:(1) SP amplitude. Does this mean only the amplitude of coupled spindles or of spindles in general?

This refers to the amplitude of spindles in general. We clarified this in the revised text (and see response to reviewer #1, point #1).

(2) The definition of a small effect

We thank the reviewer again for raising this important question. As we responded in the public review, small effect sizes are common in neuroscience and meta-analyses due to the complexity of the underlying mechanisms and the presence of numerous confounding variables and hierarchical levels. To help readers better interpret effect sizes, we changed rigid ranges to widely accepted benchmarks for effect size levels in neuroscience research: small (r=0.1), moderate (r=0.3), and large (r=0.5; Cohen, 1988). We also noted that an evidence and context-based framework will provide a more practical way to interpret the observed effect sizes compared to rigid categorizations.

(3) Can a BF10 based on experimental evidence actually be "infinite" and a probability actually be 1.00?

We appreciate the reviewer for highlighting this potential confusion. The formula used to calculate BF10 is P(data | H1) / P(data | H0). In the experimental setting with an informative prior, an ‘infinite’ BF10 value indicates that all posterior samples are overwhelmingly compatible with H1 given the data and assumptions (Cox et al., 2023; Heck et al., 2023; Ly et al., 2016). In such cases, the denominator P(data | H0) becomes vanishingly small, leading BF10 to converge to infinity. This scenario occurs when the probability of H1 converges to 1 (e.g., 0.9999999999…).

It is a well-established convention in Bayesian statistics to report the Bayes factor as "infinity" in cases where the evidence is overwhelmingly strong, and BF10 exceeds the numerical limits of the computation tools to become effectively infinite. To address this ambiguity, we added a footnote in the revised version of the manuscript to clarify the interpretation of an 'infinite' BF10 . (*p*. 8)

(4) Z_r should be renamed to r_z or similar. These are not Z values (-inf..+inf), but r values (-1..1).

We thank the reviewers for their suggestions. We agree that r_z would provide a clearer and more accurate interpretation, while z is more appropriate for referring to Fisher's z-transformed r (see point (5)). We have updated the notation accordingly.

(5) Also, it remains quite unclear at which points in the analyses, "r" values or "Fisher's z transformed r" values are used. Assumptions of normality should only apply to the transformed values. However, the formulas for the random effects model seem to assume normality for r values.

The correlation values were z-transformed during preprocessing to ensure normality and the correct estimation of sampling variances before running the models. The outputs were then back-transformed to raw r values only when reporting the results to help readers interpret the effect size. We mentioned this in Section 5.5.1, therefore the normality assumptions are not a concern. We have updated the notation r to z (-inf..+inf) in the formula of the random and mixed effect models in the revised version of the manuscript (*p*. 22).

Language(1) Frequency. In the introduction, the authors use "frequency" when they mean something like the incidence of spindles.

We agree that the term "frequency" has been used inconsistently to describe both the incidence of events and the frequency bands of oscillations. We have replaced "frequency" with "prevalence" to refer to the incidence of coupling events where applicable (*p*. 3).

(2) Moderate and mediate. These two terms are usually meant to indicate two different types of causal influences.

Thanks for the reviewer’s suggestions. We agree that "moderate" is more appropriate to describe moderators in this study since it does not directly imply causality. We have replaced mediate with moderate in relevant contexts.

(3) "the moderate effect of memory task is relatively weak": "moderator effect" or "moderate effect"?

We appreciate the reviewer for pointing out this mistake. We have updated the term to "moderator effect" in Section 2.2.2 (*p*. 6).

(4) "in frontal regions we found a latest coupled but most precise and strong SO-fast SP coupling" Meaning?

We thank the reviewer for bringing this concern of clarity to our attention. By 'latest,' we refer to the delayed phase of SO-fast SP coupling observed in the frontal regions compared to the central and parietal regions (see Figure 5), "Precise and strong" describes the high precision and strength of phase-locking between the SO up-state and the fast SP peak in these regions. We have rephrased this sentence to be: “We found that SO-fast SP coupling in the frontal region occurred at the latest phase observed across all regions, characterized by the highest precision and strength of phase-locking.” to improve clarity (*p*. 9).

(5) Figure 5 and others contain angles in degrees and radians.

We appreciate the reviewer pointing out this inconsistency. We have updated the manuscript and supplementary material to consistently use radians throughout.